# Fungal parasites infecting $N_2$-fixing cyanobacteria reshape carbon and $N_2$ fixation and trophic transfer

Anna Feuring [1], Connor D. Lawrence [1], Jessica Salcedo [1], Martin J. Whitehouse [2], Angela Vogts [1], Luca Zoccarato [3,4] & Isabell Klawonn [1] ✉

Fungal parasites are associated with bloom-forming algae, yet their impact on $N_2$ fixation and the fate of newly fixed nitrogen during cyanobacterial blooms is poorly understood. We report infections on the ecologically important $N_2$-fixing cyanobacterium *Dolichospermum* (formerly *Anabaena*) in the Baltic Sea. Using single-cell isotope probing, microscopy, and biogeochemical analyses, we examine how infections affect carbon and $N_2$ fixation and elemental transfer within a natural community. Fungal sporangia infect up to 80% of filaments, mostly targeting storage cells (akinetes, 82% prevalence) and $N_2$-fixing cells (heterocytes, 44%), but rarely vegetative cells (5%). Infections at akinete–heterocyte junctions extract 4- and 10-fold more carbon and nitrogen than those on vegetative cells, reducing host storage by 28% and 56%. Overall, 22% of newly fixed nitrogen is transferred to fungi, comparable to heterotrophic bacteria. Infections also occur in *Nodularia* and *Aphanizomenon*, suggesting fungi-like parasitism broadly affects bloom dynamics and the fate of new nitrogen.

Algal blooms have significant impacts on aquatic ecosystems and human activities, with both positive and negative consequences. On the one hand, they account for nearly half of global primary production, forming the base of aquatic food webs and supporting fisheries[1–3]. On the other hand, they can cause oxygen depletion, toxin release, and water quality degradation, threatening ecosystem functioning, biodiversity, fisheries, and recreational activities[4,5]. Under current climate change, phytoplankton blooms are expanding and intensifying in both coastal and freshwater systems[6,7], with harmful effects often outweighing the benefits. For example, in the Baltic Sea, blooms of $N_2$-fixing, partially toxin-producing cyanobacteria have increased in frequency, biomass, and duration since the 1970s[8–10]. Due to their ability to fix atmospheric $N_2$, diazotrophic cyanobacteria thrive in nitrogen-depleted waters, introducing hundreds of kilotons (180–430 kt) of new nitrogen per year into the central Baltic Sea[11,12]. This internal nitrogen loading is nearly equivalent to the external nitrogen load from river discharge (480 kt N yr$^{-1}$)[11], exacerbating the vicious cycle that fuels further algal growth, oxygen depletion, and eutrophication[13,14].

The integration of algal blooms into food webs is typically linked to interactions with zooplankton grazers, heterotrophic bacteria, protists, and viruses[15], while cyanobacterial blooms are commonly considered to be regulated by temperature and nutrient availability[16,17]. In contrast, the role of fungi—particularly parasitic fungi—has largely been overlooked. Mounting evidence, however, suggests that parasitic fungi, belonging mostly to the order Chytridiomycota (hereafter referred to as chytrids), are widespread in coastal[18–20] and freshwater ecosystems[21,22]. While thriving on major phytoplankton groups, including diatoms, cyanobacteria, and dinoflagellates, those fungal parasites infect up to 90% of the phytoplankton host population[23–25]. Parasitism, such as fungal parasitism, can thereby alter carbon transfer

[1]Department of Biological Oceanography, Leibniz Institute for Baltic Sea Research, Rostock/Warnemuende, Germany. [2]Department of Geosciences, Swedish Museum of Natural History, Stockholm, Sweden. [3]Core Facility Bioinformatics, BOKU University, Vienna, Austria. [4]Institute of Computational Biology, BOKU University, Vienna, Austria. ✉e-mail: klawonn@iow.de

pathways and food web dynamics[26,27], whereas its quantitative impact on both $N_2$ fixation and the fate of newly fixed nitrogen during cyanobacterial blooms remains unresolved.

To address this gap, we investigate the exciting interactions between parasitic fungi and $N_2$-fixing cyanobacteria, focusing on infection mechanisms and the quantitative impact on $N_2$ fixation and nitrogen transfer. Using the Baltic Sea as a study site, we apply stable-isotope tracing ($^{13}C$-bicarbonate and $^{15}N$-dinitrogen), single-cell resolution secondary-ion mass spectrometry (SIMS), partial 18S rRNA-gene sequencing, environmental monitoring, microscopy, and biogeochemical analyses. Specifically, we aim to (1) assess the prevalence of fungal infections during summer cyanobacterial blooms, particularly in *Dolichospermum* (formerly *Anabaena*); (2) quantify the transfer of carbon and nitrogen from *Dolichospermum* cells to their fungal sporangia; and (3) evaluate the impact of fungal parasitism on *Dolichospermum* populations, considering filament integrity and single-cell activity rates.

## Results

### *Dolichospermum* long-term observations

The prominent growth season of $N_2$-fixing *Dolichospermum* spp. in the Southern Baltic Sea extended from June to September between 2012 and 2021 (week 24–40, Fig. 1a). Annual peaks in *Dolichospermum* biomass predominantly occurred in July (1.9–44.8 µg C L$^{-1}$, N = 6) and, less frequently, in June (18.7 µg C L$^{-1}$, N = 1), August (2.2 and 2.8 µg C L$^{-1}$, N = 2), and September (3.3 µg C L$^{-1}$, N = 1). Between June and

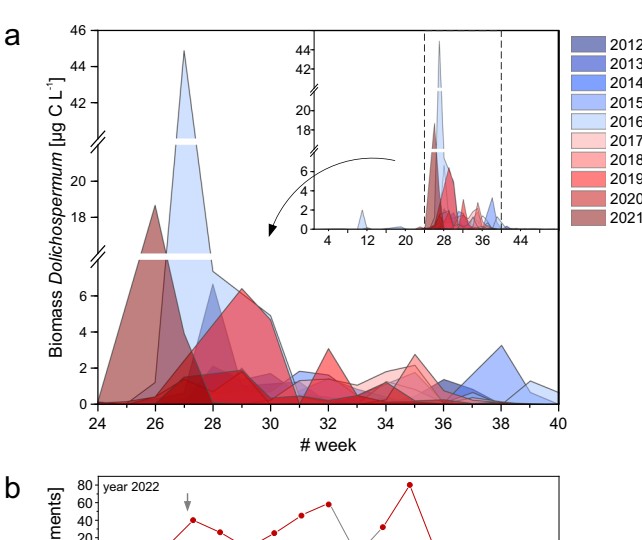

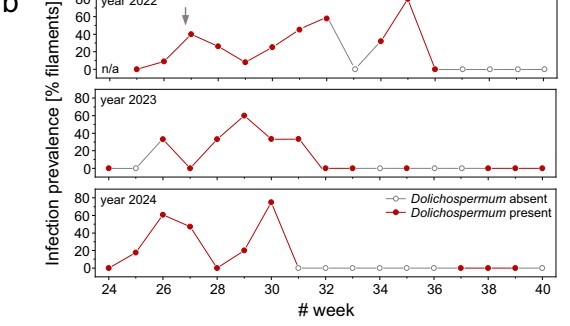

**Fig. 1 | Long-term carbon biomass and infection prevalence in *Dolichospermum* populations. a** Carbon biomass and **b** infection prevalence within $N_2$-fixing *Dolichospermum* populations at the coastal sampling station in the Southern Baltic Sea. **a** includes an inset showing biomass data over the entire year (weeks 1–52, 2012–2021). Data source: Leibniz Institute for Baltic Sea Research Warnemuende, Germany (ODIN 2 database). **b** Infection prevalence from 2022 to 2024. Weeks in which *Dolichospermum* was absent from the sampled water are marked with gray-outlined circles. The gray arrow in the top panel (year 2022) indicates the sampling date for stable isotope incubations. N/A–not analyzed due to the absence of a sample. Source data are provided as a Source Data file.

September of 2022–2024, *Dolichospermum* was detected in 33 out of 52 weekly samples. Of these 33 samples, more than half (18 samples) showed signs of infection in the *Dolichospermum* populations, with the maximum infection prevalence reaching 80% of *Dolichospermum* filaments (Fig. 1b).

### Environmental conditions during isotope-tracer incubations

Environmental conditions at the sampling site on 04 July 2022 reflected typical summer bloom conditions, with low nutrient concentrations (0.23 ± 0.06 µmol PO$_4^{3-}$ L$^{-1}$, <0.2 µmol NO$_3^-$ L$^{-1}$, <0.05 µmol NO$_2^-$ L$^{-1}$, and 0.65 ± 0.15 µmol NH$_4^+$ L$^{-1}$, N = 4 incubation bottles, Supplementary Table S1) and substantial biomass of filamentous $N_2$-fixing cyanobacteria. Specifically, *Dolichospermum* accounted for 2.0 ± 0.4 µmol C L$^{-1}$, contributing 34% to the total phytoplankton carbon biomass, while other $N_2$-fixing cyanobacteria contributed less than 1%. Non-$N_2$-fixing phytoplankton species included diatoms (44% of the phytoplankton carbon biomass), dinoflagellates (15%), picocyanobacteria (5%), and other taxa (1%, N = 4 incubation bottles, see Supplementary Table S2 for further details).

### Characteristics of fungal-infected *Dolichospermum*

Commonly, *Dolichospermum* forms chains with differentiated cell types, including vegetative cells (performing oxygenic photosynthesis), heterocytes (performing $N_2$ fixation), and akinetes (resting, energy-reserving cells). To account for this cell differentiation, we considered these three cell types separately during our analyses, as presented in the following.

The *Dolichospermum* population comprised 88 ± 13 filaments mL$^{-1}$ or 2298 ± 270 cells mL$^{-1}$, with the majority of cells being vegetative (91 ± 1%, carbon-fixing cells), followed by heterocytes (7 ± 1%, $N_2$-fixing cells) and akinetes (2 ± 1%, dormant, storage cells). Fungal infections were present in 57 ± 6% of *Dolichospermum* filaments and 9 ± 3% of their cells (N = 12 incubation bottles, Supplementary Table S3). The fungal sporangia were predominantly located at the septal junctions connecting akinetes and heterocytes (Fig. 2a). In total, 82 ± 12% of akinetes and 44 ± 8% of heterocytes were infected, whereas only 5 ± 3% of vegetative cells were infected (N = 12 incubations bottles, Fig. 2b). Rhizoids that infected akinetes and heterocytes remained restricted to those cells, whereas those growing into vegetative cells sometimes extended into adjacent vegetative cells. Fungal-infected filaments were, on average, more than twice as long as non-infected filaments (infected: 272 ± 206 µm, N = 300; non-infected: 99 ± 70 µm, N = 60; Mann–Whitney test, W = 138033.5, p = 7.68 × 10$^{-28}$, rank-biserial correlation = 0.43, 95% CI: 0.36–0.49; Fig. 2c). Moreover, akinetes were absent from 95 ± 4% of the shorter, non-infected filaments but from only 36 ± 15% of the infected filaments (N = 12 incubation bottles; Mann–Whitney test: W = 144, p = 3.21 × 10$^{-5}$, rank-biserial correlation = 1.00, 95% CI: 1.00–1.00). Consequently, the relative abundance of akinetes was significantly lower in non-infected filaments compared to infected filaments (0.5 ± 0.5% vs. 2.1 ± 0.7% of total cells, N = 12; Mann–Whitney test: W = 5, p = 1.21 × 10$^{-4}$, rank-biserial correlation = 0.93, 95% CI: 0.83–0.97).

### Bacterial colonization of non-infected and fungal-infected *Dolichospermum*

Non-infected *Dolichospermum* cells were sparsely colonized by bacteria (0.2 ± 0.6 cell$^{-1}$, N = 100), whereas fungal-infected cells exhibited roughly eight-fold higher colonization (1.7 ± 3.4 cell$^{-1}$, N = 100; ZINB model: count component: z = −4.05, p = 5.16 × 10$^{-5}$, rate ratio = 0.11, 95% CI: 0.04–0.32; Fig. 2d). This increased bacterial colonization was also observed in non-infected cells directly adjacent to fungal-infected cells, where bacterial abundance averaged 0.7 ± 2.4 bacteria cell$^{-1}$, representing an approximately three-fold increase compared with more distant non-infected cells (N = 100). A zero-inflated negative binomial (ZINB) model supported this difference, indicating a

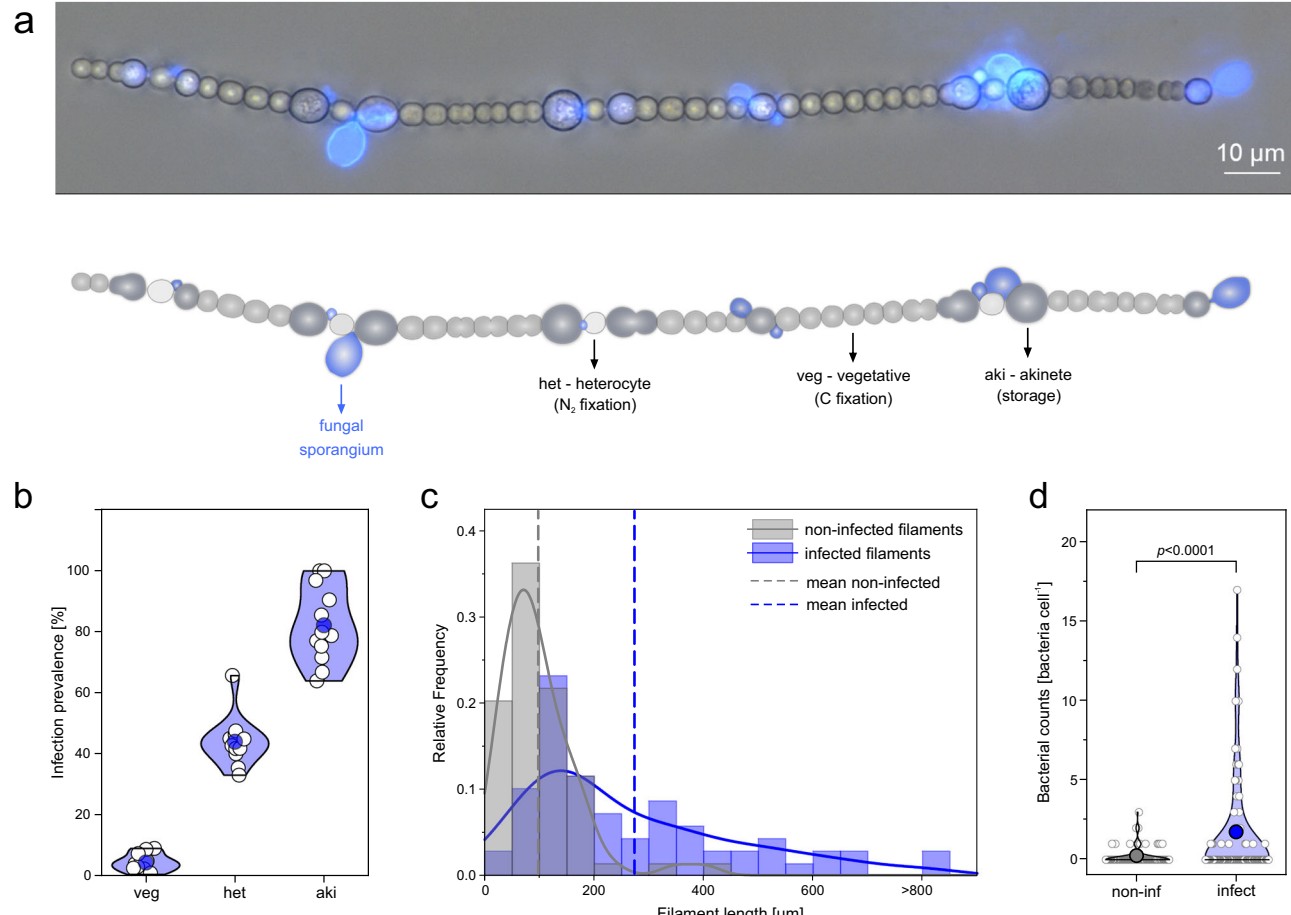

**Fig. 2 | Characteristic of fungal-infected *Dolichospermum* filaments.**
**a** *Dolichospermum* filament with fungal infections primarily located at the septal junctions between akinetes and heterocytes, while vegetative cells remained largely uninfected. Fungal sporangia are stained blue using Calcofluor White. The micrograph shows a representative *Dolichospermum* filament from the field sample collected on 04 July 2022. Incubation experiments were performed once on this sample, with N = 4 independent incubation bottles, all yielding similar results.
**b** Infection prevalence within the three differentiated cell types: vegetative cells (veg), heterocytes (het), and akinetes (aki). **c** Length distribution of non-infected and fungal-infected *Dolichospermum* filaments, shown as relative frequency (N = 65 filaments for each group). Mean values are indicated by dashed lines. Filaments longer than 800 μm (N = 2, 900 and 940 μm) are grouped as >800 μm. Infected filaments included those with mature and collapsed sporangia (excluding those with recently encysted zoospores). **d** Number of bacteria associated with individual *Dolichospermum* cells, differentiated by non-infected (non-inf) and fungal-infected cells (infect), N = 100 cells for each group (ZINB model: count component: z = −4.05, p = 5.16 × 10⁻⁵, rate ratio = 0.11, 95% CI: 0.04–0.32, two-sided). Data in **b**, **d** are presented as individual data points (white circles, N = 12 incubation bottles) and mean values (color-filled circles). **b**–**d** Distribution curves are based on kernel smoothing. Source data are provided as a Source Data file.

significantly higher expected bacterial load in adjacent cells (count component: z = 3.90, p = 9.68 × 10⁻⁵, rate ratio = 9.49, 95% CI: 3.06–29.42). The abundance of free-living bacteria was $1.13 \pm 0.14 \times 10^6$ cells mL⁻¹ (N = 12 incubation bottles).

## Fungal parasite taxonomy and life cycle

Based on 18S rRNA gene sequencing, the fungal parasite was identified as belonging to the phylum Chytridiomycota. On average, 40 ± 2% of the sequence reads were attributed to Fungi of which 96.4 ± 0.5% were identified as Chytridiomycota (N = 4 incubation bottles, Supplementary Fig. S1), which are known to parasitize a broad range of phytoplankton, including, e.g., diatoms, cyanobacteria, and dinoflagellates[28–30]. The life cycle was typical for early-branching, zoosporic fungi. Zoospores attached to and encysted on the photosynthetic *Dolichospermum* host. Following encystment, the fungus penetrated the cyanobacterial host and proliferated, conveying its nutrients to the maturing sporangium outside the host. Within this sporangium, new zoospores (ca. 10–20 zoospores) were produced. Eventually, the sporangium opened, releasing the newly generated zoospores, which were capable of swimming and infecting new host

cells. The former host cell was left with an empty, collapsed sporangium (Fig. 3a–c).

## Single-cell ¹³C and ¹⁵N₂ fixation and transfer

To comprehensively cover the multi-stage life cycle, we analyzed host cells exhibiting the three key stages of sporangial development: (1) zoospore encystment (zoospore loses the flagellum, secretes a cell wall, and adheres to the host cells), (2) mature sporangium, and (3) empty, collapsed sporangium (Fig. 3a–c). Furthermore, we categorized the host cells into distinct types, including vegetative cells, type I heterocytes (adjacent to at least one akinete), type I akinetes (adjacent to one heterocyte), type II heterocytes (enclosed by two vegetative cells), and type II akinetes (enclosed by two vegetative cells) (Fig. 3d).

Our single-cell isotope analyses addressed four central questions: Does the fungal parasite acquire more cyanobacterial-derived carbon and nitrogen when infecting septal junctions between type I akinetes and heterocytes, compared to less preferential sites such as vegetative cells, type II akinetes, and type II heterocytes? How efficiently does the fungal parasite siphon carbon and nitrogen from its cyanobacterial host? Is this cell-to-cell transfer comparable to that of bacteria? Are

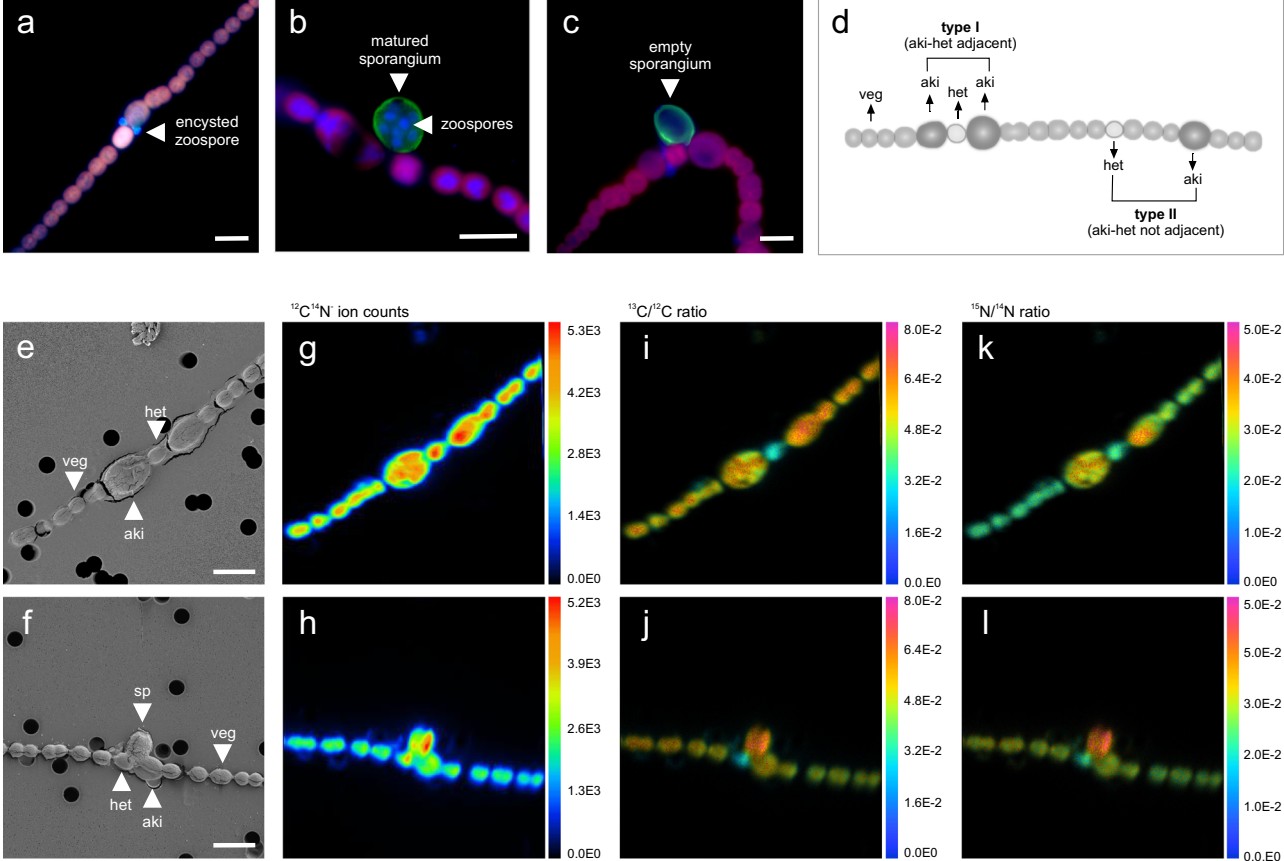

**Fig. 3 | Cell types, SEM, and SIMS images.** Sporangial development stages, including encysted zoospores (**a**), which develop into mature sporangia with multiple new zoospores inside (**b** blue-stained, roundish cells). The zoospores are discharged into the ambient water, leaving behind the cell wall remains of an empty sporangium (**c**). Cells were imaged under a fluorescence microscope after staining with Wheat Germ Agglutinin (green, chitinous cell walls) and DAPI (blue, DNA). The autofluorescence of the cyanobacterial photosynthetic pigments is displayed as dark magenta. **d** Host cell type I and II, as differentiated during single-cell isotope analyses. **e–l** Non-infected and fungal-infected *Dolichospermum* filaments (top and bottom panel, respectively) imaged under a secondary electron microscope

(**e, f** SEM, 800X magnification) and secondary-ion mass spectrometer (**g–l**, IMS1280, 70 × 70 μm raster). White scale bars are 10 μm. **g–l** were created in WinImage using the Composition/HSI function to visualize pixels with significant statistics while reducing background noise (i.e., pixels with low statistics; intensity normalization was based on the maximum numerator and denominator). Sp–fungal sporangium, veg–vegetative cell, het–heterocyte, aki–akinete). **a–c, e–l** Images show representative *Dolichospermum* filaments from the field sample collected on 04 July 2022. Incubation experiments were performed once on this sample, with N = 4 independent incubation bottles, all yielding similar results.

cyanobacterial cells of the same cell type (vegetative, heterocytes, or akinetes) with high carbon and $N_2$-fixation activity preferentially targeted for infection by the fungal parasite? Does fungal infection reduce carbon and $N_2$-fixation rates and storage in *Dolichospermum* cells?

## Site-specific acquisition of cyanobacterial carbon and nitrogen by the fungus

The fungal parasite exhibited the highest [13]C and [15]N enrichment in mature sporangia infecting type I akinete–heterocyte junctions. Specifically, the [13]C and [15]N enrichment was 4-fold and 10-fold greater in these sporangia (1.42 ± 0.33 [13]C and 0.25 ± 0.10 [15]N atom% excess, N = 33) compared to sporangia infecting vegetative cells (0.32 ± 0.33 [13]C and 0.02 ± 0.01 [15]N atom% excess, N = 41, Supplementary Fig. S2). Interestingly, sporangia infecting type II heterocytes or akinetes–which were not adjacent to each other but enclosed by vegetative cells–showed low enrichment levels, comparable to vegetative cells (type II heterocytes: 0.35 ± 0.54 [13]C and 0.04 ± 0.05 [15]N atom% excess, N = 8, type II akinetes: 0.19 ± 0.09 [13]C and 0.02 ± 0.01 [15]N atom% excess, N = 5, Supplementary Fig. S2).

Pairwise comparisons revealed that sporangia at akinete–heterocyte junctions were enriched in both [13]C and [15]N to levels exceeding those of their host cells (Fig. 4). Specifically, the [13]C

atom% excess in mature sporangia was 1.2-fold (1.2 ± 0.3, N = 24) and 4.0-fold (4.0 ± 2.1, N = 28) greater, and the [15]N atom% excess was 2.3-fold (2.3 ± 0.5, N = 24) and 3.6-fold (3.6 ± 1.5, N = 28) greater than in their type I akinete and heterocyte host cells, respectively (Fig. 4a, b). In contrast, sporangia associated with vegetative cells were significantly less enriched in both isotopes than their host cells (paired Wilcoxon signed-rank test: V = 1485, p = 1.67 × 10[-10], matched rank-biserial correlation = −1, 95% CI: −1 to −1; Fig. 4a, b). Their [13]C and [15]N enrichment was only 0.2-fold (0.21 ± 0.21) and 0.1-fold (0.14 ± 0.05, N = 54) the enrichment found in their vegetative host. Carbon and nitrogen were, therefore, efficiently transferred from type I akinetes and heterocytes to the fungal parasite, even exceeding the newly fixed carbon and nitrogen retained within the host cells (Fig. 4c, d). Conversely, newly fixed carbon and nitrogen were poorly transferred from vegetative cells to parasitizing sporangia.

[13]C and [15]N enrichment levels were similar among bacteria associated with different *Dolichospermum* host cell types: those associated with akinetes showed 0.67 ± 0.42 [13]C and 0.075 ± 0.075 [15]N atom% excess (N = 12), with heterocytes, 0.48 ± 0.44 [13]C and 0.069 ± 0.048 [15]N (N = 9), with sporangia, 0.55 ± 0.39 [13]C and 0.070 ± 0.045 [15]N (N = 38), and with vegetative cells, 0.62 ± 0.44 [13]C and 0.095 ± 0.082 [15]N (N = 63). On average, enrichments across all *Dolichospermum*-associated bacteria were 0.59 ± 0.42 [13]C atom% excess and 0.08 ± 0.07 [15]N

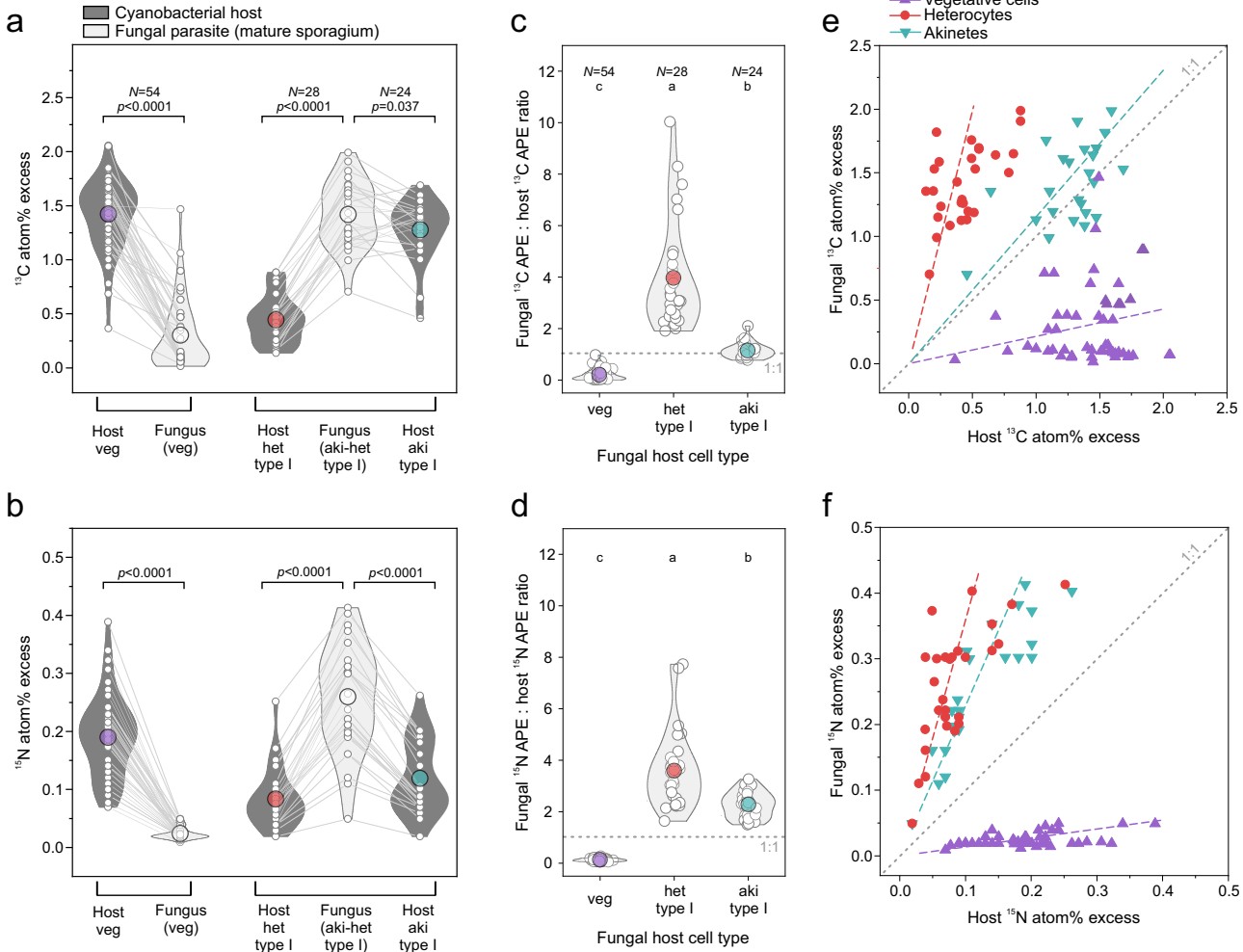

**Fig. 4 | Carbon and nitrogen transfer from cyanobacterial host to fungal parasite. a, b** Isotope enrichment ($^{13}C$ and $^{15}N$ atom percent excess) in cyanobacterial host cells and their infecting fungal sporangia, with paired data shown by connecting lines. P-values derived from paired Wilcoxon signed-rank test (two-sided, N = number of analyzed cells; $^{13}C$ at% excess: veg/veg: V = 1485, p = 1.67 × 10$^{-10}$, matched rank-biserial correlation = 1.00, 95% CI: 1.00 to 1.00; het/aki-het: V = 0, p = 4.00 × 10$^{-6}$, matched rank-biserial correlation = −1.00, 95% CI: −1.00 to −1.00; aki-het/aki: V = 77, p = 0.0383, matched rank-biserial correlation = −0.49, 95% CI: −0.76 to −0.07; and $^{15}N$ at% excess: veg/veg: V = 1485, p = 1.67 × 10$^{-10}$, matched rank-biserial correlation = 1.00, 95% CI: 1.00 to 1.00; het/aki-het: V = 0, p = 4.00 × 10$^{-6}$, matched rank-biserial correlation = −1.00, 95% CI: −1.00 to −1.00; aki-het/aki: V = 0, p = 1.94 × 10$^{-5}$, matched rank-biserial correlation = −1.00, 95% CI: −1.00 to −1.00) Ratios of $^{13}C$ APE (**c**) or $^{15}N$ APE (atom percent excess) (**d**) within the fungal parasite and their respective cyanobacterial host cells. Letters a−c denote

significantly different groups (Kruskal–Wallis test, two-sided: $^{13}C$: H(2) = 87.20, p = 1.16 × 10$^{-19}$, η²_H = 0.804, 95% CI: 0.73–0.85, with post-hoc comparisons indicating het > aki > veg, Bonferroni-corrected; $^{15}N$: H(2) = 82.38, p = 1.29 × 10$^{-18}$, η²_h = 0.76, 95% CI: 0.70−0.80, with post-hoc comparisons (Bonferroni) indicating het > aki > veg; N = number of analyzed cells). Correlations between $^{13}C$ APE (**e**) and $^{15}N$ APE (**f**) in the fungal parasite and its respective cyanobacterial host cell. Dashed trendlines represent the mean $^{13}C$ or $^{15}N$ APE ratio, as shown in (**c**) and (**d**), used to calculate the slope (y = mean ratio × host APE, intercept set to (0, 0)). **a–d** Single data points are shown as white circles. Color-filled circles represent the mean. Due to the rarity of mature sporangia located on type II heterocytes and akinetes (each enclosed by vegetative cells) in our samples, their data are not included in this figure (but see Supplementary Table S4 and Fig. S2). Source data are provided as a Source Data file.

atom% excess (N = 122). This was at least 2.4-fold lower than in mature sporangia infecting type I akinete–heterocyte junctions, but at least 1.7-fold higher than in sporangia infecting vegetative cells (Supplementary Table S4). In comparison, free-living bacteria showed considerably lower enrichment, with 0.18 ± 0.11 $^{13}C$ and 0.043 ± 0.018 $^{15}N$ atom% excess (N = 11).

Cells within the same host type (vegetative, heterocytes, or akinetes) showed no significant differences in carbon and N$_2$ fixation-based growth rates between non-infected host cells and cells that had just been infected, as indicated by an encysted zoospore (p > 0.05, two-sample Student's *t* test and Mann–Whitney, see Supplementary Fig. S3 for details). Thus, there was no preference for infection based on the level of host activity within the same cell type.

## Impact of fungal infection on cyanobacterial carbon and N$_2$ fixation

Fungal infection significantly reduced net carbon incorporation (retention) rates by 28% in type I akinetes (from 4.5 ± 1.1 pmol C cell$^{-1}$ d$^{-1}$, N = 104 to 3.2 ± 1.2 pmol C cell$^{-1}$ d$^{-1}$, N = 65; two-sided t-test: t(167) = −7.08, p = 3.76 × 10$^{-11}$, Cohen's d = −1.12, 95% CI: −1.45 to −0.79) and by 23% in type I heterocytes (from 0.10 ± 0.05 pmol C cell$^{-1}$ d$^{-1}$, N = 55 to 0.08 ± 0.04 pmol C cell$^{-1}$ d$^{-1}$, N = 62; two-sided Wilcoxon rank-sum test: W = 1232.5, p = 0.00994, rank-biserial correlation = −0.28, 95% CI: −0.46 to −0.07). The effect on vegetative cells, however, was insignificant (non-infected: 0.39 ± 0.08 pmol C cell$^{-1}$ d$^{-1}$, N = 3527 and fungal-infected: 0.38 ± 0.10 pmol C cell$^{-1}$ d$^{-1}$, N = 67; two-sided Wilcoxon rank-sum test: W = 10 746.5, p = 0.244, rank-biserial

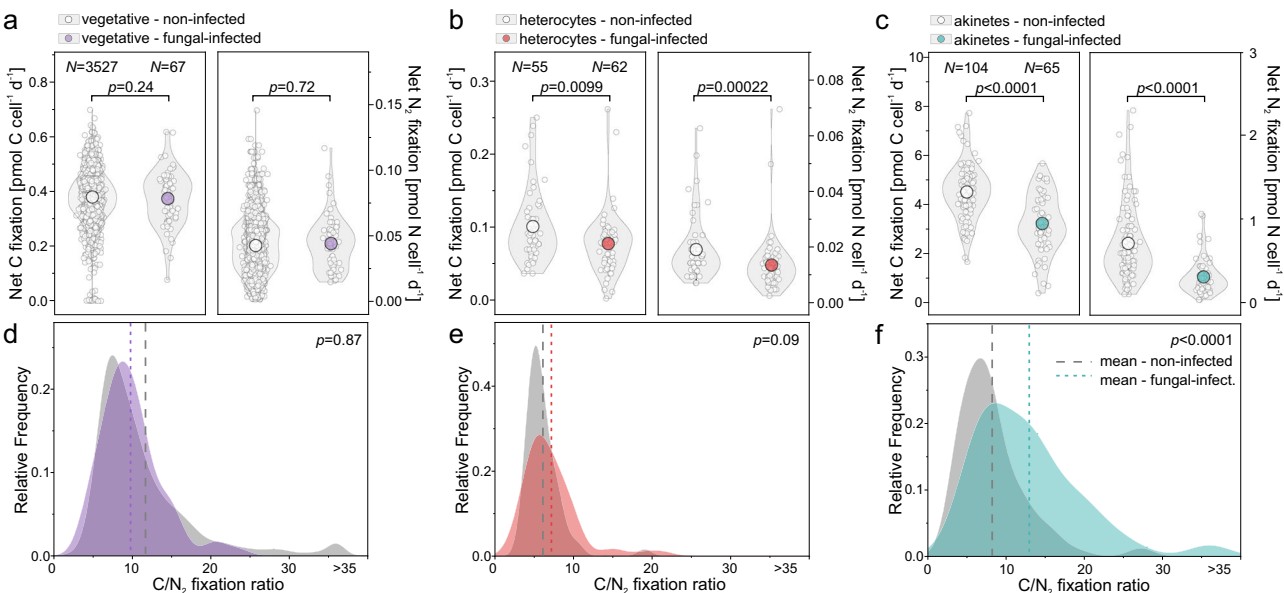

**Fig. 5 | Carbon and N$_2$ fixation-based incorporation in non-infected and fungal-infected *Dolichospermum* cells. a–c** Carbon and N$_2$ fixation-based incorporation rates in non-infected and fungal-infected *Dolichospermum* cells. Single data points are represented as white circles, with color-filled circles indicating mean values. N denotes the number of analyzed cells. Note the different axis scales. **d–f** Net C/N$_2$ incorporation ratios in *Dolichospermum* cells, shown as the relative frequency for non-infected and fungal-infected cells, with mean values outlined as dashed (non-infected) and dotted (infected) lines. Ratios greater than 35 are binned as >35. **a–f** Net carbon or N$_2$ fixation (incorporation) rates represent the amount of newly fixed carbon and nitrogen retained in host cells after some was siphoned off by the parasite or lost through other pathways during the incubation. Fungal-infected cells represent those with empty, collapsed sporangia. Statistics: **a** C-fix: Two-sided Wilcoxon rank-sum test: W = 10 746.5, p = 0.244, rank-biserial correlation = −0.24,

95% CI: −0.57 to 0.16; N-fix: Two-sided Wilcoxon rank-sum test: W = 13 073, p = 0.720, rank-biserial correlation = −0.07, 95% CI: −0.44 to 0.32. **b** C-fix: Two-sided Wilcoxon rank-sum test: W = 1232.5, p = 0.00994, rank-biserial correlation = −0.28, 95% CI: −0.46 to −0.07; N-fix: Two-sided Wilcoxon rank-sum test: W = 1030, p = 2.24 × 10$^{-4}$, rank-biserial correlation = −0.40, 95% CI: −0.56 to −0.21. **c** C-fix: Two-sided t-test: t(167) = −7.08, p = 3.76 × 10$^{-11}$, Cohen's d = −1.12, 95% CI: −1.45 to −0.79; N-fix: Two-sided Wilcoxon rank-sum test: W = 1154, p = 6.36 × 10$^{-13}$, rank-biserial correlation = −0.66, 95% CI: −0.75 to −0.54. **d** Two-sided Wilcoxon rank-sum test: W = 14 583.5, p = 0.869, rank-biserial correlation = 0.03, 95% CI: −0.35 to 0.41. **e** Two-sided Wilcoxon rank-sum test: W = 2015.5, p = 0.0905, rank-biserial correlation = 0.18, 95% CI: −0.03 to 0.38. **f** Two-sided Wilcoxon rank-sum test: W = 4942, p = 4.51 × 10$^{-7}$, rank-biserial correlation = 0.46, 95% CI: 0.31 to 0.59. Distribution curves follow kernel smoothing. Source data are provided as a Source Data file.

correlation = −0.24, 95% CI: −0.57 to 0.16, Fig. 5a–c). Similarly, net nitrogen incorporation was reduced by 56% in type I akinetes (from 0.71 ± 0.45 pmol N cell$^{-1}$ d$^{-1}$, N = 104 to 0.31 ± 0.21 pmol N cell$^{-1}$ d$^{-1}$, N = 65; two-sided Wilcoxon rank-sum test: W = 1154, p = 6.36 × 10$^{-13}$, rank-biserial correlation = −0.66, 95% CI: −0.75 to −0.54) and by 30% in type I heterocytes (from 0.018 ± 0.011 pmol N cell$^{-1}$ d$^{-1}$, N = 55 to 0.013 ± 0.010 pmol N cell$^{-1}$ d$^{-1}$, N = 62; two-sided Wilcoxon rank-sum test: W = 1030, p = 2.24 × 10$^{-4}$, rank-biserial correlation = −0.40, 95% CI: −0.56 to −0.21), while no significant change was evident in vegetative cells (non-infected: 0.04 ± 0.02 pmol N cell$^{-1}$ d$^{-1}$, N = 3527 and fungal-infected: 0.04 ± 0.02 pmol N cell$^{-1}$ d$^{-1}$, N = 67; two-sided Wilcoxon rank-sum test: W = 13 073, p = 0.720, rank-biserial correlation = −0.07, 95% CI: −0.44 to 0.32). The net C:N incorporation ratio (mol: mol) increased 1.6-fold in infected type I akinetes (from 8.3 ± 4.4, N = 104 to 12.8 ± 6.8, N = 65; two-sided Wilcoxon rank-sum test: W = 4942, p = 4.51 × 10$^{-7}$, rank-biserial correlation = 0.46, 95% CI: 0.31 to 0.59), but remained unchanged in type I heterocytes (non-infected: 6.1 ± 2.3, N = 55, fungal-infected: 7.1 ± 3.5, N = 62; two-sided Wilcoxon rank-sum test: W = 2015.5, p = 0.0905, rank-biserial correlation = 0.18, 95% CI: −0.03 to 0.38) and vegetative cells (non-infected: 11.1 ± 6.3, N = 3527, fungal-infected: 9.8 ± 3.9, N = 67; two-sided Wilcoxon rank-sum test: W = 14 583.5, p = 0.869, rank-biserial correlation = 0.03, 95% CI: −0.35 to 0.41, Fig. 5d–f).

## Discussion

The capacity of coastal and freshwater ecosystems to provide socio-economic and ecological services continues to be compromised by human activities and climate change, with algal blooms—growing both

in intensity and severity—being a major concern[7,31]. The Baltic Sea, specifically, is a prime example of eutrophic waters, challenged by large-scale cyanobacterial blooms that are partly toxic, promote internal nutrient load, and enhance oxygen depletion in both deep and shallow areas[14,32–34]. In fact, the Baltic Sea has been suggested as a natural time machine for studying the impacts and potential mitigation of future coastal disturbances, owing to its long history in multi-stressor perturbation and environmental monitoring[35]. Despite these long-term monitoring efforts, the fungal parasites reported here, which infect prominent N$_2$-fixing cyanobacteria, have gone unnoticed. Over three consecutive years, we observed fungal infections in half of our samples when *Dolichospermum* was present, with infections affecting up to 80% of filaments. Moreover, we detected fungal parasites not only on *Dolichospermum*, but also on *Nodularia* and *Aphanizomenon* in the Baltic Sea (Fig. 6, Supplementary Note 1 and Fig. S4). This suggests that fungal infections are not rare, but rather a common feature of cyanobacterial blooms in the Baltic Sea.

Akinetes, which are specialized dormant storage cells that facilitate survival, reproduction, and resilience, were preferentially infected by the fungus. For example, in our incubated population, over 80% of the akinetes in more than 50% of the filaments exhibited infections. These infection rates are notably higher than those previously reported for freshwater *Dolichospermum* (formerly *Anabaena*), where infections were observed in 4–42% of akinetes and 2–21% of filaments[25,36–39]. In contrast, the prevalence of infections in vegetative cells remained low, not exceeding 5% during our one-day incubation and throughout three consecutive years of monitoring. In Lake Aydat, Rasconi, et al[24]. observed chytrid infections preferentially targeting

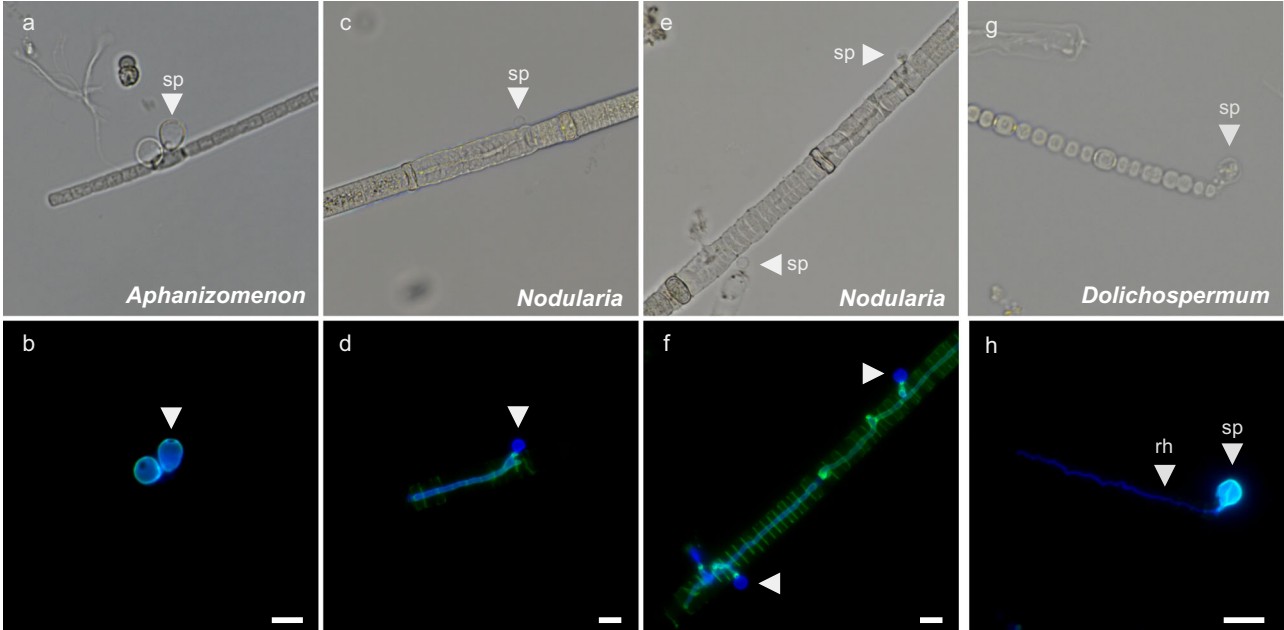

**Fig. 6 | Micrographs of fungus-like infections on the N$_2$-fixing cyanobacteria *Aphanizomenon*, *Nodularia*, and *Dolichospermum* in the Baltic Sea.** Shown are representative filaments from field samples collected in summers 2022–2024: (**a**, **c**, **e**, **g**) brightfield and (**b**, **d**, **f**, **h**) fluorescence overlays of CFW (blue) and WGA (green), both staining chitinous fungal structures, and imaged under an inverted fluorescence microscope. Rh–rhizoids; Sp–sporangia. Scale bars, 10 μm. Infection prevalence is reported in Supplementary Fig. S4.

vegetative cells, with rhizoids from a single sporangium extending through multiple cells along *Anabaena* filaments. This allowed one sporangium to infect several adjacent cells, resulting in a high infection prevalence of up to 98% of vegetative cells. We observed similar rhizoids extending along *Dolichospermum* filaments in the Baltic Sea (Fig. 6g, h), but such instances were rare. Consequently, infections of akinetes predominated over those of vegetative cells in the Baltic Sea. Interestingly, chytrids infecting either vegetative cells or akinetes have been described as distinct taxa based on morphological differences– *Rhizosiphon anabaena* and *R. akinetum* targeting akinetes, and *R. crassum* targeting vegetative cells[38,40]. Our partial 18S rRNA-gene analysis indicated the taxonomic classification within the class *Chytridiomycetes*, while higher taxonomic levels remained uncertain. Given the characteristic sporangium morphology and host cell preference, it is possible that different species coexisted on *Dolichospermum* filaments in our coastal samples, but molecular confirmation, using full-length 18S, ITS, 28S and/or whole genome sequencing for final taxonomy placement and species differentiation, is still missing. Intriguingly, the 18S rRNA gene sequence of a zoosporic fungus infecting freshwater *Dolichospermum* akinetes and indicated as *R. akinetum* (access. no. OL869110 in ref. 22) shared only 76–92% identity with our V7/8 region ASVs, suggesting a novel taxon in our sample (Supplementary Fig. S5).

Our study also provides direct evidence of fungal or fungi-like infections in *Nodularia* and *Aphanizomenon*. In both species, infections were preferentially located on heterocytes, while in *Nodularia*, the parasite extended across multiple cells within the filament (Fig. 6a–f). In the latter case, up to 71% of *Nodularia* filaments were infected (Supplementary Note 1, Supplementary Fig. S4). Similar infections have been reported recently for the Northern Baltic Sea, identifying the *Aphanizomenon* parasite as a chytrid fungus, and the *Nodularia* parasite as a fungus-like oomycete[41]. Consequently, all three major N$_2$-fixing cyanobacteria in the Baltic Sea are affected; yet, the biogeochemical imprint during blooms of *Nodularia* and *Aphanizomenon* still requires further investigation.

Filamentous cyanobacteria are often considered poorly edible due to three factors: (1) their long filaments, which most zooplankton grazers cannot consume efficiently, (2) the production of toxins that render them unpalatable to consumers, and (3) their relatively low nutritional value[42]. Nevertheless, fungal infections can make cyanobacterial blooms less of a trophic dead end[43,44]. Infections cause host filaments to break into shorter, more palatable fragments[38,45] (Supplementary Fig. S6). Additionally, nutritionally poor and partly toxic cyanobacterial biomass is converted into fungal zoospores that are rich in sterols and polyunsaturated fatty acids, providing a valuable food source for microplankton[45–48]. In this study, we observed that filaments with infections were more than twice as long (ca. 270 μm) as filaments without infections (ca. 100 μm). We hypothesize that this observation is explained by a mechanical stress within infected cells as fungi penetrate into the cyanobacterial cell cytoplasm[40]. This stress may impair cell–cell junctions, especially akinete–vegetative cell junctions, leading to filament fragmentation (Supplementary Fig. S6). This hypothesis is supported by our observations that the majority of infected filaments (60%) showed apical infections as potential fragmentation sites (see Fig. 2a for an example).

Akinete differentiation in cyanobacteria involves the transformation of vegetative cells into nutrient-storing cells. During this process, akinetes increase in volume by approximately 10-fold[49], accumulating carbon and nitrogen reserves such as glycogen and cyanophycin. The total carbon and nitrogen biomass of a single akinete is therefore equivalent to that of ten vegetative cells[50,51] (Supplementary Table S6). Given these substantial energy reserves, zoosporic fungi that specialized in parasitizing *Dolichospermum* akinetes were hypothesized to have a quantitative advantage over those infecting vegetative, carbon-fixing cells[38]. And indeed, we show that the transfer of carbon and nitrogen to sporangia infecting akinete–heterocyte junctions was 4- and 10-fold higher, respectively, than to sporangia infecting vegetative cells–despite similar carbon and N$_2$ fixation-based growth rates in both akinetes and vegetative cells (Supplementary Table S4). The $^{13}$C and $^{15}$N isotope enrichment in type I akinete sporangia even exceeded

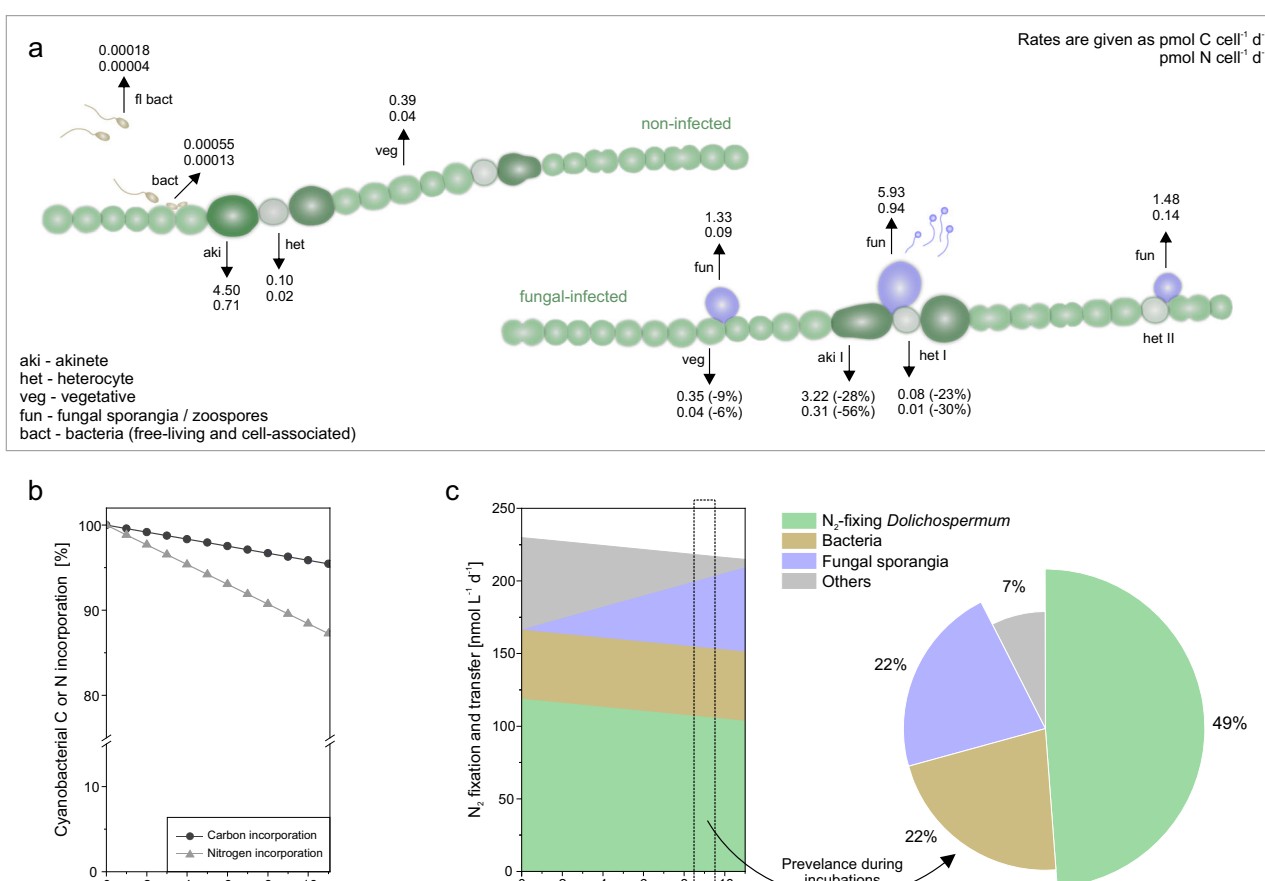

**Fig. 7 | Shifts in carbon and N₂ fixation-based incorporation rates and nitrogen transfer during fungal epidemics. a** Single-cell perspective on the incorporation and transfer rates of newly fixed carbon and nitrogen. **b** Extrapolated cyanobacterial carbon and N₂ fixation-based incorporation rates during fungal epidemics, with infection prevalence ranging from 0–11%. The cyanobacterial incorporation represents the retention of newly fixed carbon and nitrogen (N₂) within the *Dolichospermum* population during periods of reduced carbon and N₂ incorporation caused by parasitic infections. Prevalence is shown for the entire *Dolichospermum* population (% of total cells), while different prevalences in vegetative cells, heterocytes, and akinetes were considered based on the herein observed host specificity (details in Supplementary Note 2). **c** The extrapolated fate of newly fixed nitrogen, primarily incorporated by N₂-fixing *Dolichospermum* and secondarily transferred to heterotrophic bacteria, fungal sporangia, and Others. Others represent the remaining fraction in the particulate organic nitrogen (PON) pool. The pie chart illustrates the nitrogen pool at 9% infection prevalence, as prevailing during our isotope-tracer incubations. Source data are provided as a Source Data file.

that of their akinete host cells (Fig. 4), suggesting that newly stored carbon and nitrogen were efficiently and rapidly extracted by the sporangia. Net carbon and nitrogen incorporation and storage in fungal-infected type I akinetes were reduced by 28% (1.28 pmol C cell⁻¹ d⁻¹) and 56% (0.40 pmol N cell⁻¹ d⁻¹), respectively (Fig. 7a)—a loss that the next-season population of *Dolichospermum* will lack during proliferation. In contrast, net carbon and nitrogen incorporation in vegetative cells remained unchanged, and vegetative-type sporangia were ineffective at extracting newly fixed carbon and nitrogen (Fig. 4), thus growing substantially slower than type I akinete–heterocyte sporangia. Consequently, sporangia associated with akinete–heterocyte junctions appeared to occupy a sweet spot, extracting carbon from the akinete and nitrogen from the heterocyte and/or akinete (Fig. 4e, f). This sweet-spot hypothesis is further supported by the observation that sporangia associated with type II akinetes or heterocytes—each enclosed by vegetative cells—were less effective at siphoning carbon and nitrogen from host cells (Supplementary Fig. S2).

Using single-cell rates combined with cell count and biomass data, we extrapolated cyanobacterial carbon and N₂ fixation and the fate of newly fixed nitrogen within the microbial food web during fungal epidemics (Fig. 7, see Supplementary Note 2, Supplementary Tables S7 and S8 for set parameters). At 11% infection prevalence, cyanobacterial carbon fixation and/or net incorporation decreased to 95%, while N₂ fixation and/or net incorporation decreased to 87%. This 11% prevalence refers to the total *Dolichospermum* cell population, with infections assumed in 100% of akinetes, 54% of heterocytes, and 6% of vegetative cells, reflecting the herein observed proportions of infected host cell types. Concurrently, the ratio of newly fixed carbon and nitrogen retained in infected *Dolichospermum* cells increased from 8.4 to 9.2. Notably, 22% of the newly fixed nitrogen was transferred to the fungal parasite population at an infection prevalence of 9%, increasing to 27% at 11% prevalence—a transfer rate comparable to that observed for bacteria (22% at 9% prevalence, Fig. 7c). Bacterial colonization increased 8-fold on infected cells— likely due to fungal enzymatic activity that degraded the cyanobacterial mucilage sheath, which normally serves as a physical and chemical barrier[52], as well as nutrient leakage from compromised host cells, both promoting the adhesion of opportunistic bacteria[26]. Free-living bacteria, however, were numerically dominant and therefore accounted for the vast majority of nitrogen uptake within the bacterial community. In detail, within the bacterial community, nitrogen transfer to cell-associated bacteria was

negligible (<1%) compared to free-living bacteria (>99%). There is no quantitative data on nitrogen transfer to fungal parasites during cyanobacterial blooms available from the literature for comparison, neither from isolates nor from field-sampled communities. But, using isotope tracing in diatom cultures, Sánchez Barranco, et al[53]. reported a 14% nitrogen transfer, and Klawonn, et al[26]. a 20% carbon transfer from diatoms to zoosporic fungi through the fungal shunt. Additionally, Rasconi, et al[54]. applied a carbon-based model to estimate that during fungal epidemics in a cyanobacterial bloom, 37% of host-derived carbon was transferred to sporangia, accounting for 50% of the microzooplankton diet.

We report on the significant yet largely overlooked role of fungal parasites in shaping the dynamics of $N_2$-fixing cyanobacterial blooms. Our findings highlight that fungal infections not only reduce the efficiency of $N_2$ fixation but also alter the fate of newly fixed nitrogen, with up to 27% being channeled to fungal parasites. The preferential infection of akinetes—critical storage and survival structures—implies a potential long-lasting impact on cyanobacterial population dynamics. By depleting akinetes of stored carbon and nitrogen, fungal parasites may reduce the resilience of filamentous cyanobacteria and thereby impact bloom recurrence. More broadly, our findings underscore the complex role of fungal parasites in modulating harmful algal blooms—not only by potentially acting as a biological control of bloom-forming cyanobacteria, but also by altering the trophic transfer of newly fixed nitrogen, thereby challenging the paradigm that cyanobacterial blooms are primarily controlled by nutrient availability and temperature, and that they serve as a trophic bottleneck in eutrophic aquatic ecosystems.

## Methods

### Sampling and stable isotope incubations
We collected water samples during summer (July 4th, 2022, 21:00, UTC + 2) at the coast of the Southern Baltic Sea (Belt Sea, Heiligendamm, 54°08'46.7"N, 11°50'36.1"E, 0.6 m sampling depth, 5 m max. water depth) using a Niskin water sampler (Single PWS 1.7 L, Hydro-Bios, Germany). Water temperature was 18.7 °C and salinity 9.5 (measured with CTD 48 SSDA Sea & Sun Technology). The samples were transported to the nearby laboratory, and incubations started two hours after sampling.

We filled the water headspace-free into twelve 1 L Duran® bottles, amended nine of the bottles with pre-dissolved $^{15}N$-$N_2$ ($^{15}N$-$N_2$ 98%, 29817-79-6, Eurisotop, Germany) and $^{13}C$-DIC (dissolved inorganic carbon, NaH$^{13}CO_3$, 98%, 372382, Sigma-Aldrich, Germany), and closed them with a membrane lid (Duran® silicone septum, PTFE coated). The isotope amendment was $3.9 \pm 0.1$ $^{13}C$ atom% excess in the DIC pool and $0.67 \pm 0.07$ $^{15}N$ atom% excess in the $N_2$ pool ($N = 9$, see below Bulk isotope analyses). Subsequent incubations lasted for 10 h in darkness (10 h), and 10 h in darkness followed by 11 hours in light (21 h) inside a growth chamber (KBW 400, BINDER, Germany) at 18 °C with 0 µE m$^{-2}$ during darkness and 40 µE m$^{-2}$ during light. At each time point, we sampled three replicate bottles with isotope enrichment and one control bottle without enrichment for the following analyses.

### Nutrients
Samples of 80 mL were filtered through glass fiber filters (Advantec™ Grade GF75, pre-combusted at 480 °C for 5 h), the filtrate was transferred into HCl-washed (pH = 4) 30 mL-HDPE vials and frozen at −20 °C. Nutrients ($PO_4^{3-}$, $NO_3^-$, $NO_2^-$, and $NH_4^+$) were measured colorimetrically according to Grasshoff et al[55]. using a Seal Analytical QuAAtro MT3B automated constant flow analyzer for seawater (lowest standards: 0.005 µM $PO_4^{3-}$, 0.2 µM $NO_3^-$, 0.05 µM $NO_2^-$, and 0.5 µM $NH_4^+$, precision 5%). Samples for dissolved organic carbon (DOC) and dissolved nitrogen (DN) were stored in 20 mL glass vials at −20 °C and analyzed through combustion catalytic oxidation (lowest standards: 25 µmol DOC L$^{-1}$ and 5 µmol DN L$^{-1}$).

### Phytoplankton abundance and infection prevalence
We preserved 50 mL samples with Lugol's solution (100 g potassium iodide and 50 g iodine in 1 L, pH=7, 50 µL added per 1 mL sample) and stored them at 4 °C. Prior to microscopy, we decolorized Lugol samples with sodium thiosulfate ($Na_2S_2O_3$, final conc. 7.6 mM, CAS#7772-98-7, Sigma Aldrich) (Pomroy 1984) and stained with Calcofluor White (5 µg mL$^{-1}$, CFW, Fluorescent Brightener 28, Sigma Aldrich) and Wheat Germ Agglutinin (5 µg mL$^{-1}$, WGA, Alexa Fluor™ 488 Conjugate, ThermoFisher Scientific Inc., MA, USA)[56]. Samples were inspected in Utermöhl chambers at 200× (Nikon, CFI Super Fluor 20×) and 600× magnification (Nikon, CFI P-Apo 60×) with an inverted epifluorescence microscope (Nikon Eclipse Ti2, Nikon Imaging Inc., Tokyo, Japan) using a long-pass DAPI filter (DAPI-50LP) for CFW signals and a FITC filter for WGA signals. We recorded (i) *Dolichospermum* cell abundances (cells mL$^{-1}$), (ii) fungal infection prevalences in *Dolichospermum* filaments (%), (iii) infection prevalences in the three different *Dolichospermum* cell types, i.e., vegetative cells, heterocytes, and akinetes, and (iv) *Dolichospermum* filament lengths (µm, cell chains with less than four cells were not counted as filaments). We additionally analyzed samples from the initial time point for (v) abundances of co-occurring autotrophs and mixotrophs (cells mL$^{-1}$) and biomass (µmol C L$^{-1}$)[57]. Since our samples were derived from the water column, we classified *Dolichospermum* as the planktonic *Dolichospermum* (in contrast to the benthic *Anabaena*)[58], in line with the classification used in the Baltic Sea monitoring (Baltic Marine Environment Protection Commission, Helsinki Commission, HELCOM).

### Bacterial abundances
Samples for bacterial abundance were fixed with paraformaldehyde (PFA, final conc. 1.5%) and filtered onto 5.0 µm (30 mL) and 0.2 µm filters (3 mL) (Whatman® Nucleopore™ Track-Etched, 25 mm) for separating *Dolichospermum*-associated and free-living bacteria, respectively. Filters were incubated with CFW (1 µg mL$^{-1}$) and SYTOX Green (final conc. 0.1 µg mL$^{-1}$, NucGreen™ Dead 488, DNA-binding) for 10−15 min in darkness. Excess stain was rinsed off during 1 min-washing steps with PBS, MilliQ, 50% EtOH, and 100% EtOH. Filters were air-dried and mounted with a 1:1 mix of Vectashield® (Antifade Mounting Medium, Vector Laboratories, USA) and CitiFluor™ (AF1, ScienceServices GmbH, Germany). Host-associated bacteria were counted manually on 200 host cells (25 non-infected and 25 infected cells per replicate bottle), as well as their two neighboring cells. Free-living bacterial abundances were counted in 156 × 156 µm areas (N = 25 per replicate bottle) using ImageJ2 (v1.54 f)[59]. Stacked images were converted to 8-bit format, and a threshold was applied (gray value 15−255) for automated particle analysis.

### Fungal community composition
To verify the taxonomy of the unknown fungal parasite, we resolved the fungal community using 18S rRNA-gene amplicon sequencing. We filtered 500 mL onto 5.0 µm PC filters (Whatman®, Nucleopore, 25 mm), which were transferred into cryovials, placed into liquid $N_2$, and then frozen at −80 °C. To detach cells from the filters, filters were vortexed with 0.5 mm beads and 650 µL CTAB (Cetyltrimethylammonium bromide) lysing buffer (Tris−HCl 100 mM, pH 8.0, NaCl 1.4 M, EDTA 20 mM, CTAB 2%) was added[60]. After the addition of 65 µL 10% SDS and 10.6 µL Proteinase K (conc: 2 mg mL$^{-1}$, 30 units mg$^{-1}$), tubes were gently shaken and placed into a heating block at 60 °C for 1 h. Tubes were thereafter exposed to a bead-beating step (FastPrep-24TM, MP Biomedicals, time: 40 s, speed 6.0 m s$^{-1}$) and then loaded with Phenol−Chloroform−Isoamyl alcohol (P:C:I, 25:24:1). We vortexed the mixture for 5 × 3 sec, mixed by hand to ensure phenol had maximum contact with proteins, and centrifuged for 10 min at 10,000 × $g$ and 4 °C. The upper phase was transferred into a new 1.5 mL tube, and the P:C:I extraction was repeated. A final cleaning was done using chloroform:isoamyl alcohol (24:1) before the samples

were incubated in isopropanol at −20 °C overnight. Tubes were centrifuged at 16,000 × g and 4 °C for 30 min. The supernatant was removed, and the pellet was washed with ice-cold 70% ethanol. Steps of centrifugation at 17,000 x g for 10 min, supernatant removal, and EtOH washing were repeated twice. The remaining EtOH was removed, and the dried pellet was reconstituted in 30 μL PCR water at 37 °C at 450 rpm for 30 min. Prior to DNA sampling and extraction, we cleaned all disposables and surfaces with 1.2% bleach and 96% EtOH, and additionally heat-cleaned (480 °C, 5 h) the glassware to minimize DNA contamination.

The 18S rRNA-encoding DNA was amplified using the primer set FF390/FR-1 (nu-SSU-1333/nu-SSU-1647, V7/8 region)[61,62]. To reduce the co-amplification of Stramenopiles, an annealing-blocking oligonucleotide was used[62]. DNA was amplified using a Master Mix (Premix Ex Taq™ DNA Polymerase Hot Start Version, TaKaRa, Japan). Each PCR program included 240 s at 94 °C, 30 cycles of 30 s at 94 °C, 60 s at 44.5 °C, and 90 s at 72 °C, and finally 300 s at 72 °C. As a positive control, we used a microbial community DNA standard (D6306, Zymo Research Europe) to reveal possible biases throughout the workflow and analysis. Duplicate samples (i.e., technical replicates) were created by adding the same DNA extract for amplification. Amplified PCR products were purified using AGENCOURT® AMPURE® XP beads (A63881, Beckman Coulter, CA, USA) and a magnetic separator. Illumina Sequencing (MiSeq, v3 600 cycles kit, 2 × 300 bp) was performed at the Deep Sequencing Facility of the DRESDEN-concept Genome Center (Dresden, Germany).

Primers were removed from the sequences with cutadapt (v2.8). Truncation, quality filtering, dereplication, pooling, and chimera removal were done with dada2 (1.34.0)[63] in RStudio (2024.12.0)[64]. For quality filtering, a maximum of two erroneous bases was allowed for forward and reverse reads, and ten base pairs were used as an overlap when merging forward and reverse reads. For taxonomic assignment of each amplicon sequence variant (ASV), we used the PR2 SSU database (pr2_version_5.1.0_SSU_dada2) as a reference. For comparison, we also used the SILVA SSU database (silva_SSUfungi_nr99_v138_2_to-Genus_trainset). Both databases classified 39% of the ASVs as *Chytridiomycota*. Moreover, SILVA assigned 99% of those reads to class-level Chytridiomycetes Incertae Sedis, while PR2 classified 3% as Rhizophydiales, 0.04% as Chytridiales, 0.04% as Lobulomycetales, and 0.02% as Polychytriales, while the remaining reads remained unclassified at the class/order level (default minBoot=50). Additional fungal reads were assigned to Ascomycota, Basidiomycota, and Zoopagomycota (1.7%, 0.2%, and 0.9% of fungal reads). Likely contaminants were detected and removed using decontam (1.26.0)[65]. Relative abundances of ASV counts were plotted after normalizing for sample depth with DESeq2 (1.46.0)[66].

## Bulk isotope analyses
We filtered 700 mL onto pre-combusted filters (Advantec™ Grade GF75 Glass Fiber Filters, pre-combustion at 480 °C for 5 h) and stored them at −20 °C. Filters were later dried at 50 °C, fumed over HCl, pelletized into tin cups, and analyzed for $^{13}$C particulate organic carbon (POC) and $^{15}$N particulate organic nitrogen (PON) using an elemental analyzer IRMS (EA-IRMS, precision ±0.2‰ for $^{13}$C and ±0.3‰ for $^{15}$N) at the Stable Isotope Facility (SIF), UC Davis (California, US). The filtrate was transferred headspace-free into 12 mL glass Exetainer® vials (#739 W, Labco Limited, Wales, UK), preserved with zinc chloride (ZnCl$_2$, final concentration 0.05% w/v, CAS 7646-85-7, Merck), and stored at room temperature for the determination of $^{13}$C-label% in the DIC pool and $^{15}$N$_2$-label% in the dissolved N$_2$ pool. The $^{13}$C-label% of DIC was quantified by trace-gas IRMS after converting DIC to CO$_2$ using 85% phosphoric acid at the Stable Isotope Facility (SIF, UC Davis). The $^{15}$N-label% of dissolved N$_2$ was analyzed using a gas concentration and purification system interfaced with an isotope-ratio mass spectrometer in the same laboratory (raw data can be found in Supplementary

Dataset 1). Additional technical details on mass spectrometer analyses at UC Davis are provided in Supplementary Note 3. Each analysis included four replicate vials (three with isotope amendment and one control without amendment). Carbon and N$_2$ fixation or incorporation rates were calculated as follows (example shown for nitrogen):

$$N_2 \text{fixation rate} = \frac{A_{PN} - A_{PN_0}}{A_N - A_{PN_0}} \times \frac{[PN]}{\Delta t} \tag{1}$$

where *APN* is the fractional $^{15}$N-enrichment in the particulate nitrogen (PN) pool following the incubation (tn), *APNO* the fractional $^{15}$N-enrichment in the particulate nitrogen (PN) pool before the incubation (t0), *AN* the fractional $^{15}$N-enrichment (atom%) of the nitrogen source pool, *PN* the concentration of PN at the end of the incubation period, and $\Delta t$ the incubation duration[67].

## Single-cell isotope analyses
Samples (30 mL) were fixed with PFA (final conc. 1.5%), collected onto 5.0 μm filters (Isopore, TMTP02500, Merck, Darmstadt, Germany), and stored at −20 °C. Before analyses, filters were cut into circa 4 × 4 mm sections, glued onto adhesive carbon tape on a 1-inch glass slide, and coated with a 30 nm gold layer (Cressington 108 Auto Sputter Coater, Watford, UK). Cells were imaged at 800× and 1200× magnification under a scanning electron microscope (SEM, Merlin VP compact, Zeiss, Germany) to identify and later relocate non-infected and infected *Dolichospermum* cells during SIMS analyses. Given a higher sample throughput, *Dolichospermum* cells and associated sporangia were analyzed using a large-geometry secondary ion mass spectrometer (IMS1280, CAMECA, Gennevilliers, France) at the NordSIMS facility (Natural History Museum, Stockholm, Sweden) following[26]. Areas of interest (80 × 80 μm) were pre-sputtered with a primary cesium ion beam (Cs$^+$, 5 nA) for 3.5 min to remove the gold coating and cell walls. Data acquisition was done on 70 × 70 μm rasters (256 × 256 px resolution) with a 50 pA Cs+ beam (spatial resolution 500–1000 nm), utilizing the synchronized secondary ion beam raster (dynamic transfer optical system, DTOS) to maintain high transmission and high mass resolution and to facilitate reconstruction of the ion image by the instrument software. The secondary ion signals of $^{12}$C$^{14}$N$^-$, $^{12}$C$^{15}$N$^-$, and $^{13}$C$^{14}$N$^-$ were recorded by a low-noise (<0.005 counts per second) ion counting electron multiplier at least 50 planes using a peak-switching routine with dwell times of 15, 76, and 31 μs px$^{-1}$ plane$^{-1}$, respectively. The mass resolution was 12,000 (M/ΔM).

The analysis of bacteria (*Dolichospermum*-associated and free-living bacteria) required a higher spatial resolution and was thus done on a NanoSIMS 50 L (50–100 nm resolution, CAMECA, Gennevilliers, France). Areas of interest (30 × 30 μm) were pre-sputtered with a primary Cs$^+$ ion beam (600 pA) for 1.5 min. Data acquisition was done on 25 × 25 μm rasters (512 × 512 px resolution) with a 1 pA Cs$^+$ beam, utilizing the synchronized secondary ion beam raster (continuous, magnetic sector/ DTOS). The secondary ion signals of $^{12}$C$^{13}$C$^-$, $^{12}$C$^{12}$C$^-$, $^{15}$N$^{12}$C$^-$, $^{14}$N$^{12}$C$^-$, and $^{31}$P$^-$ were recorded by a low-noise (0.541 counts per second) ion counting electron current pre-amplifier over 60 planes in multi-collection mode with a dwell time of 250 μs px$^{-1}$ plane$^{-1}$. The resolution of each mass was adjusted toward the best peak identification before each measurement. The mass resolution was 10,000 M/ΔM for $^{12}$C$^{12}$C$^-$, $^{15}$N$^{12}$C$^-$, and $^{14}$N$^{12}$C$^-$ (given by CAMECA software).

Regions of interest (ROI) were selected from previously taken SEM images and drawn manually on the $^{12}$C$^{14}$N ion images using the software WinImage (v4.8). Free-living bacteria were defined as those located at least 10 μm distant to *Dolichospermum* cells (equal to approximately twice the *Dolichospermum* diameter). Isotope ratios for each ROI were calculated based on the cumulative ion counts (cts) of all planes as $^{13}$C/$^{12}$C = cts $^{13}$C$^{14}$N / cts $^{12}$C$^{14}$N$^-$ (IMS 1280), $^{13}$C/$^{12}$C = cts $^{12}$C$^{13}$C$^-$ / cts $^{12}$C$_2$×0.5 (NanoSIMS), and $^{15}$N/$^{14}$N = cts $^{15}$N$^{12}$C$^-$ / cts $^{14}$N$^{12}$C$^-$ (IMS 1280 and NanoSIMS). Isotope ratios with a standard error

exceeding twice their Poisson statistics error ($\sigma_{ratio}$), and with a relative standard deviation above 25% across all planes, were rejected. Ratios were corrected for instrument mass fractionation using our EA-IRMS data obtained from GF75 control filters (see above), following[68] and converted to atom% excess (APE), i.e., excess label% in a given cell type relative to non-enriched control cells. We used isotope atom% excess values to calculate C- and N-specific assimilation rates ($d^{-1}$) of host cells, and multiplied those by *Dolichospermum* cell-specific C- and N-contents (pmol cell$^{-1}$) (Supplementary Table S6), respectively, to obtain cell-specific rates (pmol cell$^{-1}$ d$^{-1}$), C- and N-specific rates ($d^{-1}$)[69,70] (see Supplementary Note 4 for equations). We refer to these rates (pmol cell$^{-1}$ d$^{-1}$) as net carbon or nitrogen incorporation rates, as they represent the amount of newly fixed carbon and nitrogen retained in host cells after some was siphoned off by the parasite or lost through other pathways during the incubation. Preparing samples for SIMS analyses is known to cause isotope dilution[71,72], which we did not correct for due to an uncertain dilution factor. Our sample preparation steps, however, were kept minor with only PFA preservation, known to cause moderate isotope dilution of approximately 10%[71]. Herein reported ratios should thus be considered conservative estimates.

### *Dolichospermum* long-term observations

To provide an overview of the long-term *Dolichospermum* biomass, we used long-term monitoring data obtained from the same sampling site from 1998 to 2021 (ODIN database, https://odin2.io-warnemuende.de/). Fungal infections are not included in this monitoring program. We thus collected additional samples weekly at the Heiligendamm site during 2022–2024 (supported by the IOW monitoring program), to monitor fungal infections in *Dolichospermum* populations. Depth-integrated samples were collected from the upper water column (0–1 m) with a vertical plankton net (10 µm mesh size, Apstein 50, Hydro-Bios GmbH, Germany), preserved with Lugol's solution (pH=7) and stored in 100 mL brown glass bottles at 4 °C. Using microscopy, we counted the relative abundance of fungal-infected and non-infected filaments, inspecting 100 *Dolichospermum* filaments (if present) as described above (see Phytoplankton abundance and infection prevalence).

### Statistical analyses and software

Statistical differences between two samples were calculated using the Mann–Whitney test for non-normally distributed data, the Welsh test for normally distributed data with non-equal variance, and the t-test for normally distributed data with equal variance. Normal distribution was verified using the Shapiro test and data variance with the F-test. Statistical differences between multiple groups were determined with Tukey's HSD test (for normally distributed data populations) and the Kruskal–Wallis test (if normal distribution was rejected, with Bonferroni correction for p-value adjustment). All statistical tests were two-sided. Count data of associated bacteria were dominated by zeros and thus modeled using a zero-inflated Poisson (ZIP) and Zero-Inflated Negative Binomial (ZINB) model. The model with the lower Akaike Information Criterion (AIC) was given preference. Tests were run in RStudio 2024.12.0 using the packages tidyverse[73], agricolae[74], pscl[75,76], and MASS[77]. Figures were created in RStudio 2024.12.0 (R version 4.4.2), Origin2022, and CorelDRAW2020. Microscope images were taken with the Nikon NIS-Elements (v. 5.21).

### Reporting summary

Further information on research design is available in the Nature Portfolio Reporting Summary linked to this article.

## Data availability

Sequence data have been deposited in ENA (European Nucleotide Archive) under project accession no. PRJEB96922. Accession numbers used to construct the phylogenetic tree in Supplementary Fig. S5 are listed in Supplementary Table S5, as accessed on the NCBI database. The raw mass spectrometer output can be found in Supplementary Dataset 1. Source data are provided with this paper.

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

## Acknowledgements

We sincerely thank Christian Burmeister for nutrient analyses, Jenny Jeschek for DOC/DN analyses, as well as Christin Fechtel and the entire IOW monitoring team for mycoplankton sampling during the Baltic Sea monitoring cruises and at Heiligendamm. We further acknowledge the valuable support of Susanne Busch during phytoplankton identification, Heejin Jeon during IMS-1280 analyses, Sascha Plewe, as well as Karoline Schulz, Armin Springer, and Marcus Frank from the Medical Biology and Electron Microscopic Centre (Rostock University Medical Center, Rostock) during SEM imaging, Annett Grüttmüller during nanoSIMS analyses, and Quentin Devresse and Stanislav Jabinski during proofreading. A.F., C.D.L., and I.K. were funded by the German Research Foundation (DFG, Emmy Noether grant KL 3332/1-1 to IK). Sequencing was funded by the DFG (KL 3332/3-1 to IK) and conducted at the DFG Research Infrastructure NGS Competence Center (DcGC, project 407482635) as part of the Next Generation Sequencing Competence Network (DFG project 423957469). NGS library preparation, data production and analyses were carried out at the DcGC Dresden-concept Genome Center, core facility of the CMCB and Technology Platform of the Dresden University of Technology (TU Dresden). The NordSIMS facility is operated under a Swedish Research Council infrastructure grant 2021-00276. This is NordSIMS publication #814. The NanoSIMS at the Leibnitz-Institute for Baltic Sea Research, Warnemünde (IOW) was funded by the German Federal Ministry of Education and Research (BMBF, grant 03F0626A). The IOW long-term observations were financially supported by the IOW, the Federal Maritime and Hydrographic Agency (BSH), and the state and federal government.

## Author contributions

I.K. acquired funding and designed research; A.F., C.D.L., J.S., M.J.W., A.V., and I.K. performed sampling and sample analyses; A.F., L.Z., and I.K. analyzed data; and A.F. and I.K. wrote the paper with input and approval from all coauthors.

## Funding

## Competing interests

The authors declare no competing interests.
