## [Transparent Peer Review file · Nature Communications]

Fungal parasites infecting N₂-fixing cyanobacteria reshape carbon and N₂ fixation and trophic transfer

Corresponding Author: Dr Isabell Klawonn

Version 0:

Reviewer comments:

Reviewer #1

(Remarks to the Author)

The manuscript of Feuring et al describes the discovery of cyanobacteria infecting fungal parasites and the trophic transfer of C and N to the fungi by the N₂-fixing cyanobacteria. To my knowledge, this is the first study to demonstrate this association and quantify its elemental cycling impact in nature. The authors observed close associations between the fungal parasites and the filamentous cyanobacteria over several years in a coastal area prone to algal blooms. They demonstrate with very nice stable isotope labeling experiments that the fungi take up N and C directly from the cyanobacteria, and have a stronger preference for attacking the heterocyst and akinetes (compared to vegetative cells). It is a great study, novel results, that are important for the fields of microbial ecology and biogeochemistry in general and should be of broad interest to the readership of Nature Communications. I only have one suggestion for an additional analysis - the authors should make a 18S rRNA gene phylogenetic tree (because they have the sequence) from their parasite that they discovered. This will show how it is related to other parasites that have already been discovered. Is it a new species? Or one that is already known? The tree will answer this important question. I have just a few other minor comments and suggestions, hopefully this helps the authors when revising the manuscript:

Comments:

lines 165-176: Since the whole 18S rRNA gene sequence has been obtained, the authors should provide a more precise taxonomic analysis. I recommend they do a phylogenetic analysis to see where these particular parasites fall within the Chytridiomycota tree, and whether it is a novel group or a group that is already known from other locations/studies.

starting on line 200: It is a bit unconventional to write a scientific results and discussion section with this Q1 , Q2, et c formatting. I suggest making these individual paragraphs with section headers.

lines 212-234: The discovery of the fungal preference for the N₂-fixing heterocyst and N-rich akinete storage cells over the vegetative cells is extremely interesting! A remarkable result indeed. This high nitrogen assimilation by the marine fungi associated with the N₂-fixing heterocysts makes me wonder, if organic nitrogen assimilation is an important part of the ecology of marine fungi in general. It reminds me of similarly high organic N assimilation activity inferred for marine fungi in the Namibian upwelling zone (doi.org/10.1038/s41396-021-01169-5). Could the authors comment on organic N assimilation more generally as relates to marine fungal ecology in the discussion? And, also - what the implications of fungal organic N assimilations might be for the marine N cycle?

lines 309-311: This is a really interesting discovery

lines 328-329: This is where the phylogenetic tree of the 18S gene from the chytrids studied here would be useful, R crass and R anabaena should also be represented in the tree. Is it the same species or different?

lines 331-332: This can be easily resolved using a simple 18S phylogenetic tree analysis.

Reviewer #2

(Remarks to the Author)

Feuring and colleagues present their study showing that fungal parasites significantly impact nitrogen cycling during cyanobacterial blooms by infecting key cell types in *Dolichospermum*, extracting substantial amounts of host-derived carbon and nitrogen. These infections divert newly fixed nitrogen to fungi, rivaling bacterial uptake, and suggest fungal parasitism is a widespread and influential factor in bloom dynamics in the Baltic Sea.

Though prevalent in aquatic ecosystems, we still only have a limited understanding of chytrid ecology and studies like this are therefore critical. The team are experts in applying sophisticated isotope techniques to study chytrid-algal interaction and this approach is used well here.

A generally well-written manuscript. I have a few points to consider.

To what extent the novelty of the system needs stating more clearly. This a new marine observation but has it been shown to some extent already in freshwater ecosystems?

To what extent is this a local phenomenon for the Baltic Sea? Could a statement be made of the wider significance of this work?

The early results on *Dolichospermum* distribution – is this not already known? This feels rudimentary.

The selection of specific cell types is fascinating and a key result. Could you observe the rhizoids inside the cells? Do they stay in the targeted cell-type or move to the next?

Encysted zoospore feels contradictory. Can you define? E.g. flagellum retracted?

The observation of more bacteria associated with infected cells. Brilliant result. Could more be explained on this?

I am not familiar with isotopic techniques used, so cannot comment on the validity of the approach.

Version 1:

Reviewer comments:

Reviewer #1

(Remarks to the Author)

The Reviewers have done a nice job addressing my comments, the only last remaining issue is that the nice figures and analyses that the provided in the rebuttal document were not added to the main text figures or supplemental for the revised manuscript. These new phylogenetic analyses (in particular the tree) should be reported in the manuscript (they could be in the supplemental). If the authors can make this change, then I would recommend publication.

Reviewer #1 (Remarks to the Author):

The manuscript of Feuring et al describes the discovery of cyanobacteria infecting fungal parasites and the trophic transfer of C and N to the fungi by the N₂-fixing cyanobacteria. To my knowledge, this is the first study to demonstrate this association and quantify its elemental cycling impact in nature. The authors observed close associations between the fungal parasites and the filamentous cyanobacteria over several years in a coastal area prone to algal blooms. They demonstrate with very nice stable isotope labeling experiments that the fungi take up N and C directly from the cyanobacteria, and have a stronger preference for attacking the heterocyst and akinetes (compared to vegetative cells). It is a great study, novel results, that are important for the fields of microbial ecology and biogeochemistry in general and should be of broad interest to the readership of Nature Communications. I only have one suggestion for an additional analysis - the authors should make a 18S rRNA gene phylogenetic tree (because they have the sequence) from their parasite that they discovered. This will show how it is related to other parasites that have already been discovered. Is it a new species? Or one that is already known? The tree will answer this important question. I have just a few other minor comments and suggestions, hopefully this helps the authors when revising the manuscript:

We thank the reviewer for the careful evaluation of our manuscript, the positive feedback, and constructive suggestions, which have helped us to improve the quality of the work. Below, we provide a detailed, point-by-point response to each comment.

Line numbers correspond to those in the final document without track changes. Additional edits were made to ensure compliance with the Nature Communications formatting guidelines (a title without punctuation, shortened abstract, etc.).

Comments:

lines 165-176: Since the whole 18S rRNA gene sequence has been obtained, the authors should provide a more precise taxonomic analysis. I recommend they do a phylogenetic analysis to see where these particular parasites fall within the Chytridiomycota tree, and whether it is a novel group or a group that is already known from other locations/studies.

Response: We thank the reviewer for this valuable suggestion. Our study did not target the full 18S rRNA gene, but specifically the V7/8 region amplified with primers FF390/FR-1 (nu-SSU-1333/nu-SSU-1647; see line 467 f.). These primers were chosen because they are well established for fungal community profiling (Banos et al 2018, Vainio and Hantula 2000), compatible with short-read sequencing, and allow the use of blocking oligonucleotides to reduce non-fungal amplification. The design of our study was therefore aimed at providing an overview of the fungal community composition in our environmental samples rather than in-depth phylogenetic resolution.

To nevertheless address the reviewer's point, we explored the phylogenetic placement of the most abundant fungal ASV (ASV_8, representing 87% of Chytridiomycota-assigned reads). A BLAST search on NCBI revealed a best hit with 99.7% identity to an uncultured eukaryote clone (HQ873384.1) (Prenafeta-Boldu et al 2014). Interestingly, this highly similar sequence was derived from a coastal dune system, but neither parasitic fungi nor phytoplankton/cyanobacteria were mentioned in the study (Prenafeta-Boldu et al 2014). Additional hits (89–91% identity) matched various uncultured Chytridiomycota and Rhizophydiales.

We further constructed trees directly on NCBI using (i) the 100 best BLAST matches and (ii) sequences of chytrids known to infect phytoplankton (Van den Wyngaert et al 2022) (Fig. 1 below). In the latter case (ii), ASV_8 clustered broadly within chytrid-related fungi, but with relatively low percent identities (84–92%) suggesting resolution at the order or family level, rather than genus or species (Table 1, Fig. 1B below). We also included a chytrid sequence

derived from infected *Dolichospermum* akinetes in a lake, similar to our sample, but this sequence showed only 90.7% identity with ASV 8 (OL869110.1 in Table 1), and 75.6–90.7% identity with all ASVs. We added this information to line 273 ff.

To our knowledge, the V7/8 region does not provide sufficient resolution for robust phylogenetic inference. Moreover, a publishable tree would require curated reference alignments beyond the objectives of this study. We thus prefer not to include a detailed phylogeny in the manuscript. However, our additional analyses, together with published data, highlight that the taxonomic term “chytrid” may be too narrow for our dataset. In the revised version, we therefore use more cautious terminology such as “fungal parasites,” “zoosporic fungi,” or “chytrid-like fungi.”

Table 1. Percent identity of ASV_8 with selected publicly available sequences of chytrids known to parasitize various phytoplankton hosts (data obtained from Van den Wyngaert et al 2022).

GenBank accession no.	Sequenced Organism	Identified Phytoplankton Host	Percent Identity
OL869111	uncultured Zygomycetales	Asterionella	91.6
OL869110	uncultured fungus	Dolichospermum (akinetete)	90.9
OL869119	uncultured Staurastromyces sp.	Staurastrum	90.9
OL869115	uncultured Rhizophydiales	Stephanodiscus	89.5
OL869117	uncultured Zygomycetales	Diatoma	89.5
OL869120	uncultured Rhizophydiales	Cosmarium	89.5
OL869112	uncultured Zygomycetales	Cyclotella	89.2
OL869113	uncultured Lobulomycetales	Fragilaria	87.9
OL869114	uncultured Lobulomycetales	Fragilaria	87.6
OL869116	uncultured Rhizophydiales	Synedra	87.0
OL869121	uncultured Rhizophydiales	Fragilaria	86.4
OL869118	uncultured Polyphagales	Yamagishiella	83.6

Figure 1. Phylogenetic trees created on NCBI using **(A)** the 100 best BLAST matches and **(B)** sequences of chytrids known to infect phytoplankton (data obtained from Van den Wyngaert et al 2022). ASV 8 (shown in red) represents the most abundant fungal ASV in our sample (87% of Chytridiomycota-assigned reads).

starting on line 200: It is a bit unconventional to write a scientific results and discussion section with this Q1, Q2, et c formatting. I suggest making these individual paragraphs with section headers.

Response: Yes, we agree. Individual paragraphs with headers are now implemented.

lines 212-234: The discovery of the fungal preference for the N₂-fixing heterocyst and N-rich akinete storage cells over the vegetative cells is extremely interesting! A remarkable result indeed. This high nitrogen assimilation by the marine fungi associated with the N₂-fixing heterocysts makes me wonder, if organic nitrogen assimilation is an important part of the ecology of marine fungi in general. It reminds me of similarly high organic N assimilation activity inferred for marine fungi in the Namibian upwelling zone (doi.org/10.1038/s41396-021-01169-5). Could the authors comment on organic N assimilation more generally as relates to marine fungal ecology in the discussion? And, also - what the implications of fungal organic N assimilations might be for the marine N cycle?

Response: Interesting thought. In filaments of *Anabaena/Dolichospermum*, N₂ is converted to NH₄⁺, and thereafter into glutamine and glutamate. Those amino acids are the principal export forms of fixed nitrogen from heterocytes to neighboring cells. Akinetes are differentiated resting cells, functioning as storage and survival structures. They accumulate organic nitrogen compounds, especially cyanophycin. The source of nitrogen in heterocytes and akinetes is thus mainly organic yet with different complexity—less complex amino acids in heterocytes and more complex cyanophycin in akinetes.

In general, marine fungi encode and express a diverse spectrum of carbohydrate- and protein-degrading enzymes in the oceanic water column (Baltar et al 2021, Breyer et al 2022). For instance, Orsi et al (2022) nicely demonstrated the direct assimilation of ¹³C-labelled org. carbon by diverse fungi in marine waters and sediments. Data on org. nitrogen assimilation, however, is scarce and also not included in the study by Orsi et al (2022). We thus believe that such discussion goes beyond the scope of our study—given that we focused on parasitic, early-diverging fungi (not covering the enormous diversity of marine fungi) and that we did not differentiate between different nitrogen forms.

To emphasize the role of fungal parasites on N cycling during cyanobacterial blooms, we added the conclusion: “We report on the significant yet largely overlooked role of fungal parasites in shaping the dynamics of N₂-fixing cyanobacterial blooms. Our findings highlight that chytrid infections not only reduce the efficiency of N₂ fixation but also alter the fate of newly fixed nitrogen, with up to 27% being channeled to fungal parasites.” (line 358 ff.)

lines 309-311: This is a really interesting discovery.

Response: Yes, we were also surprised that fungal infections on Baltic Sea cyanobacteria have gone mostly unnoticed so far, given that the Baltic Sea is one of the best monitored and studied water bodies globally. We believe that this discovery will change our understanding of control mechanisms, trophic transfer, and biogeochemical cycles during algal blooms.

lines 328-329: This is where the phylogenetic tree of the 18S gene from the chytrids studied here would be useful, *R. crass* and *R. anabaena* should also be represented in the tree. Is it the same species or different? lines 331-332: This can be easily resolved using a simple 18S phylogenetic tree analysis.

Response: We thank the reviewer for this comment. In our dataset, 97% of fungal reads belonged to the class Chytridiomycetes (Fig. 2A below), and ASV_8 alone represented the vast majority of those reads (Fig. 2B). Moreover, most of the remaining chytrid-related ASVs

shared at least 99% identity with ASV_8 (Table 2). At this level of sequence similarity, it seems not possible to distinguish different species with confidence.

This limitation arises because the sequenced marker—the V7/8 region of the 18S rRNA gene—offers typically genus- to family-level resolution in early-diverging fungi. Many distinct chytrid species share identical or nearly identical V7/8 sequences, and consequently, phylogenetic trees based on this region alone cannot separate species-level lineages. Robust discrimination among chytrid species generally requires additional loci (e.g., full-length 18S, 28S, and/or ITS) rather than short amplicons. We added this information to line 269 ff.

Since our V7/8 sequences shared at most 92% identity with the 18S rRNA gene sequence of a zoosporic fungus infecting freshwater *Dolichospermum* akinetes, indicated as *R. akinetum* (accession no. OL869110 in Van den Wyngaert et al (2022)), we infer that the Baltic Sea taxa are distinctly different from those occurring in lakes (as now stated in line 273 ff.). Very recently, sequences of zoosporic fungi infecting phytoplankton have also been published from the Baltic Sea, published August 7, 2025 (Van den Wyngaert et al 2025), but the corresponding 18S sequences were not yet available in NCBI (last checked August 26, 2025).

Figure 2. (A) 18S-based community composition. **(B)** Number of reads of Chytridiomycota-assigned ASVs, derived from our environmental Baltic Sea sample.

Table 2. Percent identity of Chytridiomycota-assigned ASVs, using ASV #8 as a reference.

#	ASV_ID	Percent Identity_to_ASV_8	Percent_reads
1	ASV_8	100	87.34
2	ASV_29	99.68	3.40
3	ASV_31	99.68	2.89
4	ASV_37	88.46	2.00
5	ASV_66	90.71	0.69
6	ASV_69	99.68	0.64
7	ASV_77	99.68	0.55
8	ASV_81	99.36	0.52
9	ASV_82	83.01	0.50
10	ASV_91	99.68	0.39
11	ASV_105	83.33	0.26
12	ASV_120	82.69	0.17
13	ASV_123	99.68	0.16
14	ASV_125	99.36	0.15
15	ASV_126	99.68	0.15
16	ASV_154	90.38	0.07
17	ASV_177	99.36	0.04
18	ASV_184	81.73	0.04
19	ASV_204	91.03	0.03

References

- Baltar, F., Zhao, Z., Herndl, G. J. (2021). Potential and expression of carbohydrate utilization by marine fungi in the global ocean. *Microbiome* **9**: 106.
- Banos, S., Lentendu, G., Kopf, A., Wubet, T., Glöckner, F. O., Reich, M. (2018). A comprehensive fungi-specific 18S rRNA gene sequence primer toolkit suited for diverse research issues and sequencing platforms. *BMC Microbiol.* **18**: 190.
- Breyer, E., Zhao, Z., Herndl, G. J., Baltar, F. (2022). Global contribution of pelagic fungi to protein degradation in the ocean. *Microbiome* **10**: 143.
- Orsi, W. D., Vuillemin, A., Coskun, Ö. K., Rodriguez, P., Oertel, Y., Niggemann, J., Mohrholz, V., Gomez-Saez, G. V. (2022). Carbon assimilating fungi from surface ocean to subseafloor revealed by coupled phylogenetic and stable isotope analysis. *The ISME Journal* **16**: 1245–1261.
- Prenafeta-Boldu, F. X., Summerbell, R. C., De Boer, W., Boschker, H. T. S., Gams, W. (2014). Biodiversity and ecology of soil fungi in a primary succession of a temperate coastal dune system. *Nova Hedwigia* **99**: 347-372.
- Vainio, E. J., Hantula, J. (2000). Direct analysis of wood-inhabiting fungi using denaturing gradient gel electrophoresis of amplified ribosomal DNA. *Mycol. Res.* **104**: 927-936.
- Van den Wyngaert, S., Ganzert, L., Seto, K., Rojas-Jimenez, K., Agha, R., Berger, S. A., Woodhouse, J., Padisak, J., Wurzbacher, C., Kagami, M., Grossart, H.-P. (2022). Seasonality of parasitic and saprotrophic zoosporic fungi: linking sequence data to ecological traits. *The ISME Journal* **16**: 2242–2254.

Reviewer #2 (Remarks to the Author):

Feuring and colleagues present their study showing that fungal parasites significantly impact nitrogen cycling during cyanobacterial blooms by infecting key cell types in *Dolichospermum*, extracting substantial amounts of host-derived carbon and nitrogen. These infections divert newly fixed nitrogen to fungi, rivaling bacterial uptake, and suggest fungal parasitism is a widespread and influential factor in bloom dynamics in the Baltic Sea.

Though prevalent in aquatic ecosystems, we still only have a limited understanding of chytrid ecology and studies like this are therefore critical. The team are experts in applying sophisticated isotope techniques to study chytrid-algal interaction and this approach is used well here.

A generally well-written manuscript. I have a few points to consider.

We thank the reviewer for the encouraging evaluation of our manuscript and helpful suggestions, which improved our manuscript. Below, we provide a detailed, point-by-point response.

Line numbers correspond to those in the final document without track changes. Additional edits were made to ensure compliance with the *Nature Communications* formatting guidelines (a title without punctuation, shortened abstract, etc.).

To what extent the novelty of the system needs stating more clearly. This a new marine observation but has it been shown to some extent already in freshwater ecosystems?

We clarify that fungal infections on pelagic cyanobacteria have been observed previously in freshwater (l. 61), and we detail those previous observations in the discussion (l 316) by comparing our observations with those made in freshwater. To our knowledge, we were the first to report fungal infections in *Dolichospermum* (formerly *Anabaena*) in a coastal system (Baltic Sea) when submitting our manuscript, but just very recently, on Aug 7th, 2025, Van den Wyngaert et al (2025) also report such infections on *Dolichospermum* in the Northern Baltic Sea (we included this reference in the revised version). The authors, however, neither focus specifically on *Dolichospermum* nor include any activity and transfer rates in their study. But their study confirms that fungal infections on phytoplankton are widespread across the Baltic Sea, underlining the high relevance of our novel, first-time quantitative data.

To what extent is this a local phenomenon for the Baltic Sea? Could a statement be made of the wider significance of this work?

Response: Fungal infections are not a local phenomenon. Infections have been reported worldwide, including freshwater and marine systems. We have included this wider significance statement in the introduction.

Line 53 ff.: Mounting evidence, however, suggests that parasitic fungi, belonging mostly to the order Chytridiomycota (hereafter referred to as chytrids), are widespread in coastal (Gutiérrez et al 2016, Hassett and Gradinger 2016, Kiliyas et al 2020) and freshwater ecosystems (Sime-Ngando 2012, Van den Wyngaert et al 2022). While thriving on major phytoplankton groups, including diatoms, cyanobacteria, and dinoflagellates, fungal parasites parasitize up to 90% of the phytoplankton host population (Gerphagnon et al 2017, Kagami et al 2006, Rasconi et al 2012)."

While we provide some exemplary references in the introduction, there are numerous additional references available from the literature, including e.g. those that report direct observations in coastal regions: (Garvetto et al 2019, Hanic et al 2009, Hassett et al 2017, Jobard et al 2010, Lepelletier et al 2014, Scholz et al 2014, Scholz et al 2016a, Scholz et al 2016b) (Cleary et al 2017), or those that observed a link between phytoplankton biomass and chytrid sequence reads (Duan et al 2018, Gao et al 2009) (Gutiérrez et al 2010, Wang et al 2018).

The early results on *Dolichospermum* distribution – is this not already known? This feels rudimentary.

Response: We assume that the “early results on *Dolichospermum* distribution” refer to the data shown in Fig. 1a.

Yes, the growth season of *Dolichospermum* lasting from mid-June to the beginning of August (with lower abundances occurring until the beginning of October) is known for the Southern Baltic Sea, given that this area is frequently monitored by the HELCOM and IOW-internal monitoring program (Kownacka et al 2020). The fact that we are here using publicly available monitoring data is stated in the legend of Fig. 1, line 647 ff., and in the method section, line 580/581. We included this data to align our own monitoring period, including fungal infections as a novel parameter, over the main growth season (as shown in Fig. 1b).

The selection of specific cell types is fascinating and a key result. Could you observe the rhizoids inside the cells? Do they stay in the targeted cell-type or move to the next?

Response: Good point. For fungal cells infecting heterocytes/akinetes, we did not observe rhizoids reaching out to the neighboring vegetative cells. We have now added this information in line 113 ff. By contrast, rhizoids infecting vegetative cells grew into the neighboring cells, often extending over several vegetative cells (also discussed in line 261 ff. and shown in Supplementary Fig. S4i, j).

Encysted zoospore feels contradictory. Can you define? E.g. flagellum retracted?

Response: For clarity, we revised our terminology as “zoospore encystment”, and specified this stage as follows: “zoospore loses the flagellum, secretes a wall, and adheres to the host cells” (lines 151 f.). A visual example is shown in Fig. 3a.

The observation of more bacteria associated with infected cells. Brilliant result. Could more be explained on this?

Response: Yes, we agree. Filamentous N₂-fixing cyanobacteria are commonly efficient in retaining newly fixed nitrogen, and they are embedded in a mucilage sheath that acts as a physical and chemical barrier, both limiting bacterial colonization. We hypothesize that those protective properties are deteriorated due to fungal attacks and related fungal enzymatic activities. We have included this information in the discussion (line 341 ff.).

I am not familiar with isotopic techniques used, so cannot comment on the validity of the approach.

Response: It may not be necessary, but we would like to add some words on the validity of our approach.

Isotope tracing at the single-cell level using secondary ion mass spectrometry (SIMS) is widely recognized as one of the most advanced approaches in environmental microbiology for quantifying microbial activity and tracing elemental transfer within tightly linked microbial assemblages (e.g., Mayali 2020, Pett-Ridge and Weber 2022). Our laboratory has long-standing expertise in this technique, developed through years of research and multiple studies (e.g., Klawonn et al 2016, Klawonn et al 2019, Klawonn et al 2021). We are therefore confident in the validity and robustness of our isotopic tracing approach.

References

- Cleary, A. C., Søreide, J. E., Freese, D., Niehoff, B., Gabrielsen, T. M. (2017). Feeding by *Calanus glacialis* in a high arctic fjord: Potential seasonal importance of alternative prey. *ICES J. Mar. Sci.* **74**: 1937-1946.
- Duan, Y., Xie, N., Song, Z., Ward, C. S., Yung, C. M., Hunt, D. E., Johnson, Z. I., Wang, G. (2018). A high-resolution time series reveals distinct seasonal patterns of planktonic fungi at a temperate coastal ocean site (Beaufort, North Carolina, USA). *Appl. Environ. Microbiol.* **84**: e00967-00918.
- Gao, Z., Johnson, Z. I., Wang, G. (2009). Molecular characterization of the spatial diversity and novel lineages of mycoplankton in Hawaiian coastal waters. *The ISME Journal* **4**: 111–120.
- Garvetto, A., Badis, Y., Perrineau, M.-M., Rad-Menéndez, C., Bresnan, E., Gachon, C. M. M. (2019). Chytrid infecting the bloom-forming marine diatom *Skeletonema* sp.: Morphology, phylogeny and distribution of a novel species within the Rhizophydiales. *Fungal Biology* **123**: 471-480.
- Gerphagnon, M., Colombet, J., Latour, D., Sime-Ngando, T. (2017). Spatial and temporal changes of parasitic chytrids of cyanobacteria. *Sci Rep.* **7**: 6056.
- Gutiérrez, M. H., Pantoja, S., Quiñones, R. A., González, R. R. (2010). First record of filamentous fungi in the coastal upwelling ecosystem off central Chile. *Gayana* **74**: 66-73.
- Gutiérrez, M. H., Jara, A. M., Pantoja, S. (2016). Fungal parasites infect marine diatoms in the upwelling ecosystem of the Humboldt current system off central Chile. *Environ. Microbiol.* **18**: 1646-1653.
- Hanic, L. A., Sekimoto, S., Bates, S. S. (2009). Oomycete and chytrid infections of the marine diatom *Pseudonitzschia pungens* (Bacillariophyceae) from Prince Edward Island. *Botany* **87**: 1096-1105.
- Hassett, B. T., Gradinger, R. (2016). Chytrids dominate arctic marine fungal communities. *Environ. Microbiol.* **18**: 2001-2009.
- Hassett, B. T., Ducluzeau, A. L. L., Collins, R. E., Gradinger, R. (2017). Spatial distribution of aquatic marine fungi across the western Arctic and sub-arctic. *Environ. Microbiol.* **19**: 475-484.
- Jobard, M., Rasconi, S., Sime-Ngando, T. (2010). Fluorescence in situ hybridization of uncultured zoospore fungi: Testing with clone-FISH and application to freshwater samples using CARD-FISH. *J. Microbiol. Methods* **83**: 236-243.
- Kagami, M., Gurung, T. B., Yoshida, T., Urabe, J. (2006). To sink or to be lysed? Contrasting fate of two large phytoplankton species in Lake Biwa. *Limnol. Oceanogr.* **51**: 2775-2786.
- Kilias, E. S., Junges, L., Šupraha, L., Leonard, G., Metfies, K., Richards, T. A. (2020). Chytrid fungi distribution and co-occurrence with diatoms correlate with sea ice melt in the Arctic Ocean. *Communications Biology* **3**: 183.
- Klawonn, I., Nahar, N., Walve, J., Andersson, B., Olofsson, M., Svedén, J. B., Littmann, S., Whitehouse, M. J., Kuypers, M. M. M., Ploug, H. (2016). Cell-specific nitrogen- and carbon-fixation of cyanobacteria in a temperate marine system (Baltic Sea). *Environ. Microbiol.* **18**: 4596-4609.
- Klawonn, I., Bonaglia, S., Whitehouse, M. J., Littmann, S., Tienken, D., Kuypers, M. M. M., Brüchert, V., Ploug, H. (2019). Untangling hidden nutrient dynamics: rapid ammonium cycling and single-cell ammonium assimilation in marine plankton communities. *The ISME Journal* **13**: 1960-1974.
- Klawonn, I., Van den Wyngaert, S., Parada, A. E., Arandia-Gorostidi, N., Whitehouse, M. J., Grossart, H.-P., Dekas, A. E. (2021). Characterizing the “fungal shunt”: Parasitic fungi on diatoms affect carbon flow and bacterial communities in aquatic microbial food webs. *Proc. Natl. Acad. Sci. USA* **118**: e2102225118.
- Kownacka, J., Busch, S., Göbel, J., Gromisz, S., Hällfors, H., Högländer, H., Huseby, S., Jaanus, A., Jakobsen, H. H., Johansen, M., Johansson, M., Jurgensone, I., Kraśniewski, W., Kremp, A., Lehtinen, S., Olenina, I., v.

- Weber, M., Wasmund, N. (2020). *HELCOM Baltic Sea Environment Fact Sheet: Cyanobacteria biomass 1990-2019*.
- Lepelletier, F., Karpov, S. A., Alacid, E., Le Panse, S., Bigeard, E., Garcés, E., Jeanthon, C., Guillou, L. (2014). *Dinomyces arenysensis* gen. et sp. nov. (Rhizophydiales, Dinomycetaceae fam. nov.), a chytrid infecting marine dinoflagellates. *Protist* **165**: 230-244.
- Mayali, X. (2020). NanoSIMS: Microscale Quantification of Biogeochemical Activity with Large-Scale Impacts. *Ann. Rev. Mar. Sci.* **12**: 449-467.
- Pett-Ridge, J., Weber, P. K. (2022). NanoSIP: NanoSIMS Applications for Microbial Biology. *Methods in molecular biology (Clifton, N.J.)* **2349**: 91–136.
- Rasconi, S., Niquil, N., Sime-Ngando, T. (2012). Phytoplankton chytridiomycosis: Community structure and infectivity of fungal parasites in aquatic ecosystems. *Environ. Microbiol.* **14**: 2151-2170.
- Scholz, B., Küpper, F. C., Vyverman, W., Karsten, U. (2014). Eukaryotic pathogens (Chytridiomycota and Oomycota) infecting marine microphytobenthic diatoms - a methodological comparison. *J. Phycol.* **50**: 1009-1019.
- Scholz, B., Guillou, L., Marano, A. V., Neuhauser, S., Sullivan, B. K., Karsten, U., Küpper, F. C., Gleason, F. H. (2016a). Zoosporic parasites infecting marine diatoms - A black box that needs to be opened. *Fungal Ecol.* **19**: 59-76.
- Scholz, B., Küpper, F. C., Vyverman, W., Karsten, U. (2016b). Effects of eukaryotic pathogens (Chytridiomycota and Oomycota) on marine benthic diatom communities in the Solthörn tidal flat (southern North Sea, Germany). *Eur. J. Phycol.* **51**: 253-269.
- Sime-Ngando, T. (2012). Phytoplankton chytridiomycosis: Fungal parasites of phytoplankton and their imprints on the food web dynamics. *Front. Microbiol.* **3**: 1-13.
- Van den Wyngaert, S., Ganzert, L., Seto, K., Rojas-Jimenez, K., Agha, R., Berger, S. A., Woodhouse, J., Padisak, J., Wurzbacher, C., Kagami, M., Grossart, H.-P. (2022). Seasonality of parasitic and saprotrophic zoosporic fungi: linking sequence data to ecological traits. *The ISME Journal* **16**: 2242–2254.
- Van den Wyngaert, S., Nawaz, A., Alacid, E., Wood-Rocca, S. M., Reñé, A., Garcés, E., Kremp, A., Wurzbacher, C. (2025). Dynamics of zoosporic parasites in summer phytoplankton communities of the Baltic Sea. *FEMS Microbiol. Ecol.* **101**.
- Wang, Y., Sen, B., He, Y., Xie, N., Wang, G. (2018). Spatiotemporal distribution and assemblages of planktonic fungi in the coastal waters of the Bohai Sea. *Front. Microbiol.* **9**: 584.

Reviewer #1 (Remarks to the Author):

The Reviewers have done a nice job addressing my comments, the only last remaining issue is that the nice figures and analyses that the provided in the rebuttal document were not added to the main text figures or supplemental for the revised manuscript. These new phylogenetic analyses (in particular the tree) should be reported in the manuscript (they could be in the supplemental). If the authors can make this change, then I would recommend publication.

We agree and have added the phylogenetic tree to the Supplementary (Supplementary Figure S5).

We wish to thank the reviewer again for the careful review.